# PLUG-AND-PLAY POSTERIOR SAMPLING UNDER MISMATCHED MEASUREMENT AND PRIOR MODELS

**Marien Renaud**[*]
Washington University in St. Louis
St. Louis, MO 63130, USA

**Jiaming Liu**
Washington University in St. Louis
St. Louis, MO 63130, USA

**Valentin de Bortoli**
ENS, CNRS, PSL University
Paris, 75005, FRANCE

**Andrés Almansa**
CNRS, Université Paris Cité
Paris, 75006, FRANCE

**Ulugbek Kamilov**
Washington University in St. Louis
St. Louis, MO 63130, USA

## ABSTRACT

Posterior sampling has been shown to be a powerful Bayesian approach for solving imaging inverse problems. The recent *plug-and-play unadjusted Langevin algorithm (PnP-ULA)* has emerged as a promising method for Monte Carlo sampling and minimum mean squared error (MMSE) estimation by combining physical measurement models with deep-learning priors specified using image denoisers. However, the intricate relationship between the sampling distribution of PnP-ULA and the mismatched data-fidelity and denoiser has not been theoretically analyzed. We address this gap by proposing a posterior-$L_2$ pseudometric and using it to quantify an explicit error bound for PnP-ULA under mismatched posterior distribution. We numerically validate our theory on several inverse problems such as sampling from Gaussian mixture models and image deblurring. Our results suggest that the sensitivity of the sampling distribution of PnP-ULA to a mismatch in the measurement model and the denoiser can be precisely characterized.

## 1 INTRODUCTION

Many imaging problems can be formulated as *inverse problems* seeking to recover high-quality images from their low-quality observations. Such problems arise across the fields of biomedical imaging (McCann et al., 2017a), computer vision (Pizlo, 2001), and computational imaging (Ongie et al., 2020). Since imaging inverse problems are generally ill-posed, it is common to apply prior models on the desired images. There has been significant progress in developing *Deep Learning* (DL) based image priors, where a deep model is trained to directly map degraded observations to images (McCann et al., 2017b; Jin et al., 2017; Li et al., 2020).

Model-based DL (MBDL) is an alternative to traditional DL that explicitly uses knowledge of the forward model by integrating DL denoisers as implicit priors into model-based optimization algorithms (Venkatakrishnan et al., 2013; Romano et al., 2017). It has been generally observed that learned denoisers are essential for achieving the state-of-the-art results in many imaging contexts (Metzler et al., 2018; Ulondu-Mendes et al., 2023; Ryu et al., 2019; Hurault et al., 2022; Wu et al., 2020). However, most prior work in the area has focused on methods that can only produce point estimates without any quantification of the reconstruction uncertainty (Belhasin et al., 2023), which can be essential in critical applications such as healthcare or security (Liu et al., 2023).

In recent years, the exploration of strategies for sampling from the posterior probability has emerged as a focal point in the field of inverse problem in imaging (Pereyra et al., 2015; Bouman & Buzzard, 2023; Chung et al., 2023; Song et al., 2022). This pursuit has given rise to a plethora of techniques, encompassing well-established methods such as Gibbs sampling (Coeurdoux et al., 2023), the Unadjusted Langevin Algorithm (ULA) (Roberts & Tweedie, 1996; Durmus et al., 2018), and more contemporary innovations like conditional diffusion models (Chung et al., 2023; Kazerouni et al., 2022; Kawar et al., 2022; Song et al., 2023), and Schrödinger bridges (Shi et al., 2022).

---

[*]Corresponding author: `marien.renaud@math.u-bordeaux.fr`

Among these sampling methods, ULA, characterized by its Markov chain-based framework, has gained prominence owing to its ease of implementation and recent versions (Cai et al., 2023; Klatzer et al., 2023; Ehrhardt et al., 2023). *Plug-and-Play-ULA* (PnP-ULA) (Laumont et al., 2022) is a specific variant that incorporates the prior knowledge into the dynamics of the Markov chain through a denoiser. While this technique stands out for its simplicity and its ability to approximate the posterior law effectively, it is not immune to challenges, including computational time and distribution shifts between the mathematical formulations and practical experiments. These distribution shifts arise due to several factors. First, there is a distribution shift in the prior distribution (Jalal et al., 2021; Chung & Ye, 2022), attributed to the approximation of the *minimum mean squared error* (MMSE) denoiser. Second, a distribution shift emerges in the data-fidelity term due to the inherent uncertainty in the forward model (Wirgin, 2004; Guerquin-Kern et al., 2012; Blanke et al., 2020). The impact of these shifts on the efficacy of the sampling method presents an intriguing gap in the current theoretical understanding.

**Contributions.** **(a)** Bayesian posterior sampling relies on two operators: a data-fidelity term and a denoising term. This paper stresses that in the case of mismatched operators, errors do not accumulate indefinitely. Moreover, with mismatched operators, the shift in the sampling distribution can be quantified by a unified formulation, as presented in Theorem 1. **(b)** Furthermore, we provide a generalized convergence result for PnP-ULA (Laumont et al., 2022) in Corollary 1.2. A new proof strategy based on Girsanov theory allows us to relax assumptions on the denoiser accuracy. These insights are substantiated by a series of experiments conducted on both a 2D Gaussian Mixture prior and image deblurring scenarios.

## 2 BACKGROUND

**Inverse Problem.** Many problems can be formulated as an inverse problem involving the estimation of an unknown vector $\boldsymbol{x} \in \mathbb{R}^d$ from its degraded observation $\boldsymbol{y} = \boldsymbol{A}\boldsymbol{x} + \boldsymbol{n}$, where $\boldsymbol{A} \in \mathbb{R}^{m \times d}$ is a measurement operator and $\boldsymbol{n} \sim \mathcal{N}(0, \sigma^2 \boldsymbol{I}_m)$ is usually the Gaussian noise.

**Posterior Sampling.** When the estimation task is ill-posed, it becomes essential to include additional assumptions on the unknown $\boldsymbol{x}$ in the estimation process. In the *Bayesian* framework, one can utilize $p(\boldsymbol{x})$ as the prior to regularize such estimation problems, and samples from the posterior distribution $p(\boldsymbol{x}|\boldsymbol{y})$. The relationship is then established formally using Bayes's rule $p(\boldsymbol{x}|\boldsymbol{y}) \propto p(\boldsymbol{y}|\boldsymbol{x})p(\boldsymbol{x})$, where $p(\boldsymbol{y}|\boldsymbol{x})$ denoted as the likelihood function.

In this paper, we focus on the task of sampling the posterior distribution based on Langevin stochastic differential equation (SDE) $(\boldsymbol{x}_t)_{t \in \mathbb{R}^+}$ (Roberts & Tweedie, 1996; Durmus et al., 2018) as

$$d\boldsymbol{x}_t = \nabla \log p(\boldsymbol{x}_t|\boldsymbol{y}) + \sqrt{2}d\boldsymbol{z}_t = \nabla \log p(\boldsymbol{y}|\boldsymbol{x}_t) + \nabla \log p(\boldsymbol{x}_t) + \sqrt{2}d\boldsymbol{z}_t, \tag{1}$$

where $(\boldsymbol{z}_t)_{t \in \mathbb{R}^+}$ is a $d$-dimensional Wiener process. A posterior sampling approach produces multiple solutions for the same degradation (more details in Appendix C.1). When $p(\boldsymbol{x}|\boldsymbol{y})$ is proper and smooth, with $\nabla \log p(\boldsymbol{x}|\boldsymbol{y})$ Lipschitz continuous, it has been proven that the stochastic process defined in equation 1 has a unique strong solution which admits the posterior $p(\boldsymbol{x}|\boldsymbol{y})$ as unique stationary distribution (Roberts & Tweedie, 1996). In practice, an Euler-Maruyama discretization of Equation 1 define the Unadjusted Langevin algorithm (ULA) Markov chain for all $k \in \mathbb{N}$ as

$$\boldsymbol{x}_{k+1} = \boldsymbol{x}_k + \delta \nabla \log p(\boldsymbol{y}|\boldsymbol{x}_k) + \delta \nabla \log p(\boldsymbol{x}_k) + \sqrt{2\delta}\boldsymbol{z}_{k+1}, \tag{2}$$

where $\boldsymbol{z}_k \sim \mathcal{N}(0, 1)$ and $\delta > 0$ is a step size controlling a trade-off between asymptotic accuracy and convergence speed (Dalalyan, 2017). The likelihood score $\nabla \log p(\boldsymbol{y}|\boldsymbol{x})$ can be computed using the measurement model[1]. However, the prior score $\nabla \log p(\boldsymbol{x})$ cannot be computed explicitly and needs to be approximate.

**Score approximation using PnP Priors.** Score approximation is a key problem in machine learning which can be solved in various ways such as Moreau-Yosida envelope (Durmus et al., 2018), normalizing flows (Cai et al., 2023) or score-matching (Nichol & Dhariwal, 2021). In this work, we approximate to the prior score $\nabla \log p(\boldsymbol{x})$ through a MMSE denoiser $D_\epsilon^\star$ denoted as

$$D_\epsilon^\star(\boldsymbol{z}) := \mathbb{E}[\boldsymbol{x}|\boldsymbol{z}] = \int_{\mathbb{R}^d} \boldsymbol{x} p_{\boldsymbol{x}|\boldsymbol{z}}(\boldsymbol{x}|\boldsymbol{z})d\boldsymbol{x}, \tag{3}$$

---

[1]In this paper, we will assume that the measurement model is known, even though it can be inexact.

where $\boldsymbol{z} = \boldsymbol{x} + \boldsymbol{n}$ with $\boldsymbol{x} \sim p(\boldsymbol{x}), \boldsymbol{n} \sim \mathcal{N}(0, \epsilon^2 \boldsymbol{I}_d), \boldsymbol{z} \sim p_\epsilon(\boldsymbol{z})$ . Since the exact MMSE denoiser $D_\epsilon^\star$ is generally intractable in practice, one can approximate it by a deep neural network (DNN) denoiser $D_\epsilon \neq D_\epsilon^\star$. This DNN denoiser is trained by minimizing the mean squared error (MSE) loss (Xu et al., 2020). It gives a link between the intractable prior distribution and the MMSE denoiser which can be approximated through Tweedie's formula (Efron, 2011)

$$\nabla \log p(\boldsymbol{x}) \approx \nabla \log p_\epsilon(\boldsymbol{x}) = \frac{1}{\epsilon} \left( D_\epsilon^\star(\boldsymbol{x}) - \boldsymbol{x} \right) \approx \frac{1}{\epsilon} \left( D_\epsilon(\boldsymbol{x}) - \boldsymbol{x} \right). \tag{4}$$

The right-hand side of equation 4 is then plugged in the ULA Markov chain to obtain the PnP-ULA

$$\boldsymbol{x}_{k+1} = \boldsymbol{x}_k + b_\epsilon(\boldsymbol{x}_k) + \sqrt{2\delta} \boldsymbol{z}_{k+1}, \tag{5}$$

where the deterministic term of the process $b_\epsilon$ corresponds to the drift function defined as

$$b_\epsilon(\boldsymbol{x}) = \nabla \log p(\boldsymbol{y}|\boldsymbol{x}) + \frac{1}{\epsilon} \left( D_\epsilon(\boldsymbol{x}) - \boldsymbol{x} \right) + \frac{1}{\lambda} \left( \Pi_{\mathbb{S}}(\boldsymbol{x}) - \boldsymbol{x} \right),$$

with $\Pi_{\mathbb{S}}$ is the orthogonal projection on the convex-compact $\mathbb{S}$. This term is added for theoretical purposes (ensures that the Markov chain is bounded) but it is rarely activated in practice. The drift $b_\epsilon(\boldsymbol{x})$ of this Markov chain (5) also corresponds the approximation of the posterior score $\nabla \log p(\boldsymbol{x}|\boldsymbol{y})$.

Despite all the approximations made by PnP-ULA, it has been shown by (Laumont et al., 2022) that, under certain relevant assumptions, this Markov chain (5) possesses a unique invariant measure $\pi_{\epsilon,\delta}$ and converges exponentially fast to it. This means that $\pi_{\epsilon,\delta}$ is the sampling distribution limit of the Markov chain (5). In practice, a significant number of step (hundred of thousands) is computed to ensure the convergence. So, we will refer to $\pi_{\epsilon,\delta}$ as the sampling distribution. This has an impact on the computational time required by PnP-ULA which exceeds that of alternative methods, such as diffusion model or flow matching model (Delbracio & Milanfar, 2023). Note that the difference between $\pi_{\epsilon,\delta}(\boldsymbol{x})$ and $p(\boldsymbol{x}|\boldsymbol{y})$ has been quantified in total variation ($TV$) distance (Laumont et al., 2022, Proposition 6).

## 3  PNP-ULA SENSITIVITY ANALYSIS

In this section we study the impact of a drift shift on the invariant distribution of the PnP-ULA Markov chain. Such a shift can be observed in both a denoiser shift or a forward model shift. Most of the introduced assumptions are reminiscent of previous work on PnP-ULA analysis (Laumont et al., 2022).

With two different drifts $b_\epsilon^1$ and $b_\epsilon^2$, we define the two corresponding Markov chains for $i \in \{1, 2\}$

$$\boldsymbol{x}_{k+1}^i = \boldsymbol{x}_k^i + \delta b_\epsilon^i(\boldsymbol{x}_k^i) + \sqrt{2\delta} \boldsymbol{z}_{k+1}^i. \tag{6}$$

Hence, the variables subject to modification in the expression of $b_\epsilon^i$ are only the forward model $\boldsymbol{A}^i$ and the denoiser $D_\epsilon^i$ :

$$b_\epsilon^i(\boldsymbol{x}) = -\frac{1}{2\sigma^2} \nabla \|\boldsymbol{y} - \boldsymbol{A}^i \boldsymbol{x}\|^2 + \frac{1}{\epsilon} \left( D_\epsilon^i(\boldsymbol{x}) - \boldsymbol{x} \right) + \frac{1}{\lambda} \left( \Pi_{\mathbb{S}}(\boldsymbol{x}) - \boldsymbol{x} \right),$$

With $\| \cdot \|$ the Euclidean norm on $\mathbb{R}^d$. The forward model $\boldsymbol{A}^i$ and the denoiser $D_\epsilon^i$ can be viewed as parameters of the PnP-ULA Markov chain. Our goal is to study the sensitivity of the sampling distribution $\pi_{\epsilon,\delta}^i$ to these parameters. For simplicity in this paper, we will designate $b_\epsilon^1$ as the reference drift and $b_\epsilon^2$ as the mismatched drift. The proposed analysis of a drift shift is based on the following assumptions.

**Assumption 1.** *The prior distributions $p^1(\boldsymbol{x})$ and $p^2(\boldsymbol{x})$, denoted as target and mismatched priors, have a finite second moment, $\forall i \in \{1, 2\}, \int_{\mathbb{R}^d} ||\boldsymbol{x}||^2 p^i(\boldsymbol{x}) d\boldsymbol{x} < +\infty.$*

This assumption is a reasonable assumption since many images have bounded pixel values, for example $[0, 255]$ or $[0, 1]$.

**Assumption 2.** *The forward model has a bounded density, $\forall \boldsymbol{y} \in \mathbb{R}^m, \sup_{\boldsymbol{x} \in \mathbb{R}^d} p(\boldsymbol{y}|\boldsymbol{x}) < +\infty$. Moreover, the forward model is smooth with Lipschitz gradient, $p(\boldsymbol{y}|.) \in \mathbf{C}^1(\mathbb{R}^d, ]0, +\infty[)$ and there exists $L > 0$ such that, $\forall \boldsymbol{y} \in \mathbb{R}^d, \nabla \log \left( p(\boldsymbol{y}|\cdot) \right)$ is $L$-lipschitz.*

Assumption 2 is true if the forward problem is linear, which will be the case in our applications.

**Assumption 3.** *There exists $\epsilon_0 > 0$, $M \geq 0$ such that for any $\epsilon \in ]0, \epsilon_0]$, $\boldsymbol{x}_1, \boldsymbol{x}_2 \in \mathbb{R}^d$ :*

$$||D_\epsilon(\boldsymbol{x}_1) - D_\epsilon(\boldsymbol{x}_2)|| \leq M||\boldsymbol{x}_1 - \boldsymbol{x}_2||.$$

Assumption 3 holds if the activation functions of the DNN denoiser are Lipschitz (e.g., ReLU). The constant $M$ is independent of $\epsilon$, when the denoisers are blind denoisers.

**Assumption 4.** *There exists $m \in \mathbb{R}$ such that for any $\boldsymbol{x}_1, \boldsymbol{x}_2 \in \mathbb{R}^d$, $\forall i \in \{1, 2\}$ :*

$$\langle \nabla \log p^i(\boldsymbol{y}|\boldsymbol{x}_2) - \nabla \log p^i(\boldsymbol{y}|\boldsymbol{x}_1), \boldsymbol{x}_2 - \boldsymbol{x}_1 \rangle \leq -m||\boldsymbol{x}_2 - \boldsymbol{x}_1||^2.$$

Note that if Assumption 4 is satisfied with $m > 0$, then the likelihood $\boldsymbol{x} \mapsto \log p(\boldsymbol{y}|\boldsymbol{x})$ is $m$-concave. If the forward model $\boldsymbol{A}$ is not invertible, such as deblurring, then $m < 0$. If Assumption 2 holds, then Assumption 4 holds with $m = -L$. However, it is possible that $m > -L$ which leads to better convergence rates for PnP-ULA. To ensure the stability of PnP-ULA in the case of $m < 0$, the projection on $\mathbb{S}$ has been added.

We introduce metrics which will be used to quantify the difference between the sampling distributions $\pi^1_{\epsilon,\delta}$ and $\pi^2_{\epsilon,\delta}$. The first one is the $TV$ distance which quantifies the point-wise distance between the densities of the probability distributions. The $TV$ distance is the distance induced by the $TV$ norm defined for a probability density $\pi$ by :

$$||\pi||_{TV} = \sup_{||f||_\infty \leq 1} \left| \int_{\mathbb{R}^d} f(\boldsymbol{x}) d\pi(\boldsymbol{x}) \right|. \tag{7}$$

The second one is the Wasserstein distance which is the cost of optimal transport from one distribution to the other (Villani et al., 2009). Formally, The Wasserstein-1 distance between two distribution $\pi_1, \pi_2$ can be defined as :

$$\mathbf{W}_1(\pi_1, \pi_2) = \inf_{\mu \in \Gamma(\pi_1, \pi_2)} \int_{\mathbb{R}^d \times \mathbb{R}^d} ||\boldsymbol{x}_1 - \boldsymbol{x}_2|| d\mu(\boldsymbol{x}_1, \boldsymbol{x}_2), \tag{8}$$

Where $\Gamma(\pi_1, \pi_2)$ denoted all transport plans having $\pi_1$ and $\pi_2$ as marginals.

## 4 MAIN RESULT

To present our theoretical analysis, we first define a new pseudometric between two functions taking values in $\mathbb{R}^d$

**Definition 4.1** (Posterior-$L_2$ pseudometric). The posterior-$L_2$ pseudometric between $f_1$ and $f_2$ : $\mathbb{R}^d \mapsto \mathbb{R}^d$ is defined by :

$$d_1(f_1, f_2) = \sqrt{\mathbb{E}_{X \sim \pi^1_{\epsilon,\delta}} \left( ||f_1(X) - f_2(X)||^2 \right)}. \tag{9}$$

The properties of the posterior-$L_2$ pseudometric are studied in Supplement A. For simplicity, we shall call it posterior-$L_2$. This metric can be computed in practice because $\pi^1_{\epsilon,\delta}$ is sampling distribution of PnP-ULA run with the drift $b^1_\epsilon$.

We can now state our main result on PnP-ULA sensitivity to a drift shift.

**Theorem 1.** Let Assumptions 1-4 hold true. There exists $A_0, B_0, A_1, B_1 \in \mathbb{R}^+$, such that for $\delta \in ]0, \bar{\delta}]$ :

$$||\pi^1_{\epsilon,\delta} - \pi^2_{\epsilon,\delta}||_{TV} \leq A_0 d_1(b^1_\epsilon, b^2_\epsilon) + B_0 \delta^{\frac{1}{4}},$$

$$\mathbf{W}_1(\pi^1_{\epsilon,\delta}, \pi^2_{\epsilon,\delta}) \leq A_1 \left( d_1(b^1_\epsilon, b^2_\epsilon) \right)^{\frac{1}{2}} + B_1 \delta^{\frac{1}{8}},$$

With $\bar{\delta} = \frac{1}{3}(L + \frac{M+1}{\epsilon} + \frac{1}{\lambda})^{-1}$. The proof is provided in Supplement D.

Theorem 1 establishes that a drift shift implies a bounded shift in the sampling distribution. Both the total variation ($TV$) and Wasserstein distance bounds consist of two different terms. The first

term depends on the posterior-$L_2$ distance $d_1(b_\epsilon^1, b_\epsilon^2)$, which represents the drift shift error. The second term is a discretization error term that depends on the step-size $\delta$. The Wasserstein distance bound seems suboptimal, since it has been derived directly from the $TV$ bound. A similar bound in Wasserstein distance, analogous to the $TV$-distance, could potentially be demonstrated. However, we leave this for future work. Constants $A_0, B_0, A_1, B_1$ have a polynomial dimension dependence (Appendix D.1.6).

The pseudometric $d_1$ also appears in the analysis of diffusion models. For example in (Chen et al., 2023; Benton et al., 2023; Conforti et al., 2023) the convergence bounds are expressed in terms of $\int_{\mathbb{R}^d} \|\nabla \log q_t(x_t) - s_\theta(t, x_t)\|^2 q_t(x_t) \mathrm{d}x_t$, where $q_t$ is the density of the forward process and $s_\theta$ is the approximated score. This quantity also corresponds to the difference of the drifts between the *ideal* backward diffusion process and the approximated backward diffusion used in practice. Similar tools, namely Girsanov theory, are used in both our analysis and the error bounds of diffusion models.

Our analysis is backward-compatible with the previous theoretical results (Laumont et al., 2022, Proposition 6) on PnP-ULA. In addition, our result provides a reformulation of (Laumont et al., 2022, Proposition 6) if denoisers $D_\epsilon^i$ are close to the exact MMSE denoiser $D_\epsilon^\star$ in the infinite norm. However, it is worth to note that our result is more general because no assumptions are made on the quality of denoiser $D_\epsilon^i$. Another key difference lies in the posterior-$L_2$ pseudometric, which is relevant to characterize the Wasserstein distance, as we can see in our experiments (see Section 5).

## 4.1 IMPLICATIONS AND CONSEQUENCES

In this section, we will present three consequences of Theorem 1 in different application cases. All these results are presented in $TV$-distance for simplicity, but can also be easily derived in Wasserstein distance. A demonstration of these results can be found in Supplement E.

### 4.1.1 DENOISER SHIFT

Theorem 1 has an interesting consequence in case of a denoiser shift. The posterior-$L_2$ appears to be a relevant metric to compare different denoisers efficiencies to provide a high-quality sampling.

**Corollary 1.1.** Let Assumptions 1-4 hold true. There exists $A_2, B_2 \in \mathbb{R}^+$, such that $\forall \delta \in ]0, \bar{\delta}]$ and for all denoisers $D_\epsilon^1$ and $D_\epsilon^2$ :

$$\|\pi_{\epsilon,\delta}^1 - \pi_{\epsilon,\delta}^2\|_{TV} \leq A_2 d_1(D_\epsilon^1, D_\epsilon^2) + B_2 \delta^{\frac{1}{4}}.$$

Similar to Theorem 1, there are two terms: the first one quantifying the denoiser shift, and the second one is the discretization error. The proof of this corollary can be found in Supplement E.1. It provides a quantification of the sensitivity of the invariant law of PnP-ULA to the denoiser, which is a key component of the process. This can be viewed as the process's sensitivity to regularization, or in a Bayesian paradigm, to prior knowledge.

### 4.1.2 REFORMULATION OF PNP-ULA CONVERGENCE GUARANTIES

Another consequence is the reformulation of PnP-ULA's previous convergence result (Laumont et al., 2022, Proposition 6), which previously required the denoiser to be close to the exact MMSE denoiser $D_\epsilon^\star$ (see equation 3). We have demonstrated a similar convergence result without this assumption.

More precisely, by naming $p_\epsilon(\cdot|\boldsymbol{y})$ the posterior distribution with $p_\epsilon$ (see equation 2) as a prior distribution, the following result holds.

**Corollary 1.2.** Let Assumptions 1-5 hold true. Let $\lambda > 0$ such that $2\lambda(L + \frac{M+1}{\epsilon} - \min(m, 0)) \leq 1$ and $\epsilon \leq \epsilon_0$. There exists $C_3 \geq 0$ such that for $R_C > 0$ such that $\bar{B}(0, R_C) \subset \mathbb{S}$ there exist $A_3, B_3, \in \mathbb{R}^+$ such that $\forall \delta \in ]0, \bar{\bar{\delta}}]$ :

$$\|p_\epsilon(\cdot|\boldsymbol{y}) - \pi_{\epsilon,\delta}^2\|_{TV} \leq A_3 d_1(D_\epsilon^\star, D_\epsilon^2) + B_3 \delta^{\frac{1}{4}} + C_3 R_C^{-1}.$$

A demonstration can be found in Supplement E.2. This result requires another technical assumption 5 which is nothing more than some regularity on the exact MMSE denoiser $D_\epsilon^\star$.

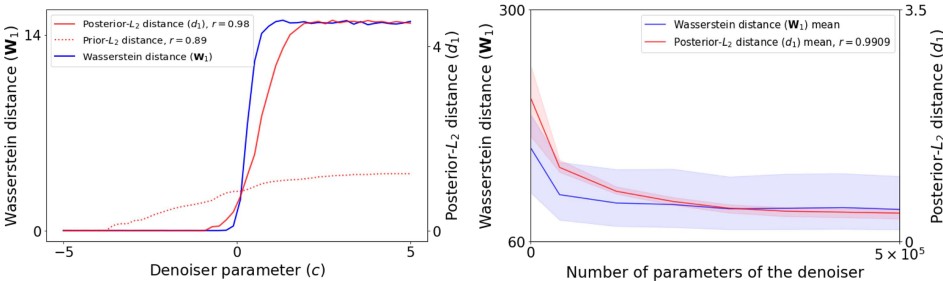

Figure 1: Illustration of Theorem 1's bound by visualizing the strong correlation between the Wasserstein distance between sampling distributions and the posterior-$L_2$ distance between denoisers. *Left plot:* Distances, for the GMM experiment in 2D, computed between sampling generated by mismatch denoisers and the exact MMSE denoiser. Note how the posterior-$L_2$ is more correlated to the Wasserstein distance than the prior-$L_2$. *Right plot:* Distances, for the gray-scale images experiment, compute between DnCNN denoisers with $5 \times 10^5$ weights and other DnCNN denoisers with fewer weights. Note how the posterior-$L_2$ and the Wasserstein distance are highly correlated with correlation $r = 0.9909$ in average and $r > 0.97$ for each image.

### 4.1.3 FORWARD MODEL SHIFT

Another consequence of Theorem 1 is in case of a forward model mismatch (Wirgin, 2004; Blanke et al., 2020), a high-stakes subject especially in medical imaging (Guerquin-Kern et al., 2012). It implies that if the shift of the forward model is limited, then the shift on the sampling distribution is limited.

**Corollary 1.3.** Let Assumptions 1-4 hold true. There exist $A_3, B_3 \in \mathbb{R}^+$, such as $\forall \delta \in ]0, \bar{\delta}]$ :

$$\|\pi^1_{\epsilon,\delta} - \pi^2_{\epsilon,\delta}\|_{TV} \le A_4\|\boldsymbol{A}^1 - \boldsymbol{A}^2\| + B_4\delta^{\frac{1}{4}}.$$

A demonstration can be found in E.3. The spectral norm is used here, but this result holds true for any norm in the matrix space because all norms are equivalent in finite-dimensional space. Thus the stability of PnP-ULA with respect with the measurement model has been proved.

## 5 NUMERICAL EXPERIMENTS

Our main result in Theorem 1 provides error bounds for the distance between two sampling distributions $\pi^1_{\epsilon,\delta}$ and $\pi^2_{\epsilon,\delta}$, as a function of the corresponding posterior-$L_2$. We provide numerical validations of this bound by exploring the behavior of the Wasserstein distance between sampling distributions and the corresponding posterior $L_2$ pseudometric defined Eq 4.1. We use the correlation between these two distances to validate our theoretical results and show that the posterior-$L_2$ pseudometric can characterize the Wasserstein distance.

The three experiments illustrate the usefulness of the three different corollaries in the previous section. Our first experiment (section 5.1) illustrates sampling distribution error (Corollary 1.2) for a GMM in 2D. The second experiment (section 5.2) illustrates denoiser shift (Corollary 1.1) for an gray-scale image deblurring. The third example in section 5.3 illustrates forward model shift (Corollary 1.3) on color image deblurring.

More results of PnP-ULA on images can be found in Supplement C. More technical details about experiments can be found in Supplement B. The code used in these experiments can be found in PnP ULA posterior law sensivity code.

### 5.1 DENOISER SHIFT ON GAUSSIAN MIXTURE MODEL IN 2D

In general, computing the exact MMSE denoiser $D^\star_\epsilon$ is a challenging task. However, if we assume that the prior distribution $p(\boldsymbol{x})$ follows a Gaussian Mixture Model (GMM), then a closed-form expression for $D^\star_\epsilon$ becomes available. This allows us to evaluate Corollary 1.2 in a simplified scenario involving a 2D GMM. We emphasize the ability of the posterior-$L_2$ pseudometric between denoisers

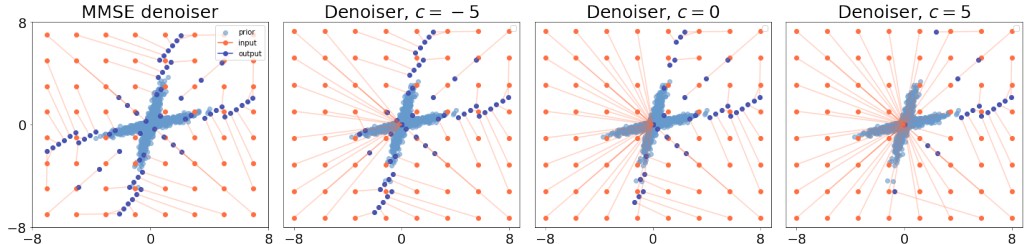

Figure 2: Illustration of denoisers with $\epsilon = 0.05$ used for the 2D Gaussian Mixture experiment. The prior distribution, a Gaussian Mixture, is represented in light blue. Denoisers, $D_\epsilon : \mathbb{R}^2 \to \mathbb{R}^2$, are represented by there outputs (in dark blue) on a set of inputs (in orange) linked together (by orange lines). *Rightmost:* Exact MMSE denoiser. *Leftmost:* Three mismatch denoisers with various $c$ parameter.

to explain variations in the Wasserstein distance between sampling distributions. Notably, denoisers are trained based on the prior distribution, which might lead one to expect that the prior-$L_2$ norm would provide a more accurate measure. However, we will illustrate that this is not the case.

The algorithm is run for a denoising problem ($A = I_2$, $\sigma^2 = 1$) and the observation $y = (0, 8)$. With PnP-ULA parameters : $\epsilon = 0.05$, $\delta = 0.05$ and $N = 100000$. The initialization of the Markov chain is taken at $x = 0$. Samples, $s^\star = (s_k^\star)_{k \in [1, N]}$, generated by the exact MMSE denoiser $D_\epsilon^\star$ provide a reference sampling distribution $\pi^\star = \frac{1}{N} \sum_{k=1}^{N} \delta_{s_k^\star}$.

We introduce a novel class of denoising operators referred to as *mismatched denoisers*, denoted as $D_\epsilon^c$ . These operators are defined to be equal to the exact MMSE denoiser when the horizontal coordinate $x_1$ exceeds a threshold $c$ and 0 otherwise.

$$D_\epsilon^c(x_1, x_2) := \begin{cases} D_\epsilon^\star(x_1, x_2) & \text{if } x_1 > c \\ 0 & \text{otherwise} \end{cases} \tag{10}$$

It becomes evident that $\lim_{c \to -\infty} D_\epsilon^c = D_\epsilon^\star$ and conversely $\lim_{c \to +\infty} D_\epsilon^c = 0$. In Figure 2, we visualize denoisers used in the experiment. A total of 50 distinct mismatched denoisers are systematically generated, spanning the parameter range $c \in [-5, 5]$. Concomitant with each of these denoisers $D_\epsilon^c$, a corresponding sample $s^c = (s_k^c)_{k \in [1, N]}$ and sampling distribution $\pi^c = \frac{1}{N} \sum_{k=1}^{N} \delta_{s_k^c}$ is generated.

In Figure 1, we present the Wasserstein distance, denoted as $\mathbf{W}_1(\pi^\star, \pi^c)$, between the reference sampling distribution $\pi^\star$ and the mismatched sampling distribution $\pi^c$. Concurrently, we compute the prior-$L_2$ pseudometric between denoisers, $\sqrt{\frac{1}{N} \sum_{k=1}^{N} \|D_\epsilon^\star(x_k) - D_\epsilon^c(x_k)\|^2}$, with $(x_k)_{k \in [1, N]}$ a sample of the prior distribution. Furthermore, we display the posterior-$L_2$ pseudometric, $d_1(D_\epsilon^\star, D_\epsilon^c) = \sqrt{\frac{1}{N} \sum_{k=1}^{N} \|D_\epsilon^\star(s_k^\star) - D_\epsilon^c(s_k^\star)\|^2}$. The relation between these metrics is explained in Corollary 1.2. Upon a closer examination of Figure 1, it becomes evident that the posterior-$L_2$ pseudometric exhibits a stronger correlation with the Wasserstein distance, $r = 0.98$, when compared to the prior-$L_2$ pseudometric, $r = 0.89$. This experiment shows the relevance of the posterior-$L_2$ pseudometric (at the basis of Theorem 1) to reflect the accuracy of a posterior sampling method.

## 5.2 DENOISER SHIFT ON GRAY-SCALE IMAGES

In practical applications, the learned denoising model does not equate to the exact MMSE denoiser. Furthermore, the testing distribution rarely aligns perfectly with the training distribution, giving rise to a distributional mismatch that can be understood as a form of *incorrect training* for the denoiser. In this context, it becomes imperative to understand the sensitivity of the invariant distribution of PnP-ULA concerning the denoiser. This sensitivity is elucidated by Equation 1.1.

In order to validate empirically this result, the deblurring task is addressed using a uniform blur kernel with dimensions of $9 \times 9$ and a noise level of $\sigma = \frac{1}{255}$. This degradation is applied to natural

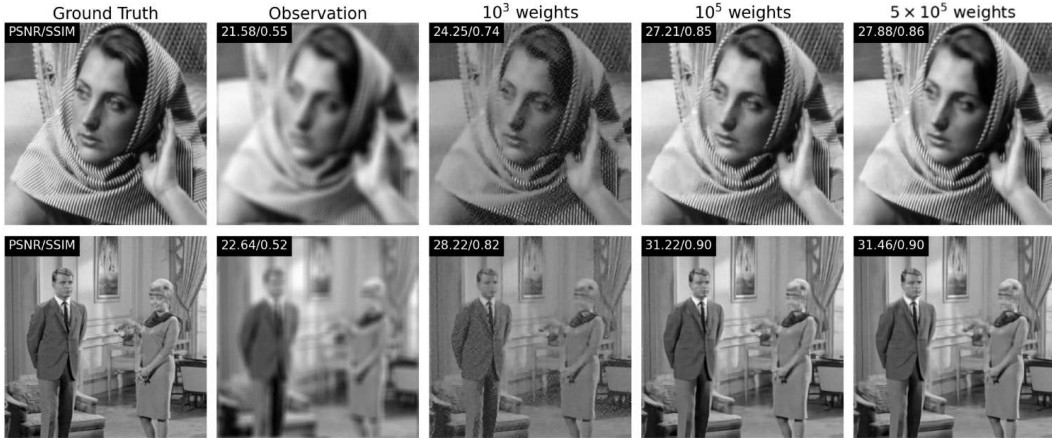

Figure 3: Illustration of MMSE estimators computed by PnP-ULA run on $10^5$ steps with various DnCNN denoisers. The quantities in the top-left corner of each image provide PSNR and SSIM values for each reconstructed images. Denoisers have a different number of weights, but are all trained in the same way. Note that a shift between the reference denoiser ($5 \times 10^5$ weights) and mismatched denoisers using less weights ($10^3$ or $10^5$ weights) implies a shift in the MMSE estimator quality.

images (from CBSD68 dataset (Martin et al., 2001)) of size $256 \times 256$ in grayscale. A total of 17 distinct DnCNN denoising models (Ryu et al., 2019) were trained, each varying in the number of layers, ranging from a single layer to a maximum of 17 layers. PnP-ULA was runned for a duration of $10^4$ steps using each of these denoising models across 15 different images (see Supplement B). Parameters are chosen following the recommendation of (Laumont et al., 2022). The initialization of PnP-ULA was performed using the observation vector $\boldsymbol{y}$, and a total of $N = 1000$ images were saved as samples, with one image being saved every 10 steps in the sampling process.

We denote by $s^i = (s_k^i)_{1 \leq k \leq N}$ the samples generated using the denoiser $D_\epsilon^i$ with $i$ layers and $\pi^i = \frac{1}{N} \sum_{k=1}^{N} \delta_{s_k^i}$ the corresponding sampling distribution. $\pi^{17}$ is our reference sampling distribution. Figure 1 illustrates the Wasserstein distance between sampling distributions denoted as $\mathbf{W}_1(\pi^{17}, \pi^i)$, in comparison to the posterior-$L_2$ between denoisers, defined as $d_1(D_\epsilon^{17}, D_\epsilon^i) = \sqrt{\frac{1}{N} \sum_{k=1}^{N} \|D_\epsilon^{17}(s_k^{17}) - D_\epsilon^i(s_k^{17})\|^2}$. Results on images are depicted in Figure 3 to provide empirical evidence that a more powerful denoiser results in a more precise reconstruction.

The figures display both the mean and the range of results across the 15 images. For each image (note that constants of Theorem 1 are problem-specific), we compute the correlation, and the average correlation is computed to be $r = 0.9909$, with the correlation between the two distances for each image $r > 0.97$.

It is important to note that the Wasserstein distance does not tend to zero because of a bias in this scenario. A sample of 1000 images in a space of dimension $256 \times 256 = 65,536$ is insufficient. Therefore, two samples from the same distribution should not be expected to have a Wasserstein distance of zero.

Evidently, the two distances exhibit a significant correlation, showing a link between these two distances. Consequently, the objective of this experiment is achieved. Furthermore, it is apparent that the samples generated by the denoiser with only $10^5$ weights closely resemble those produced by the denoiser $5 \times 10^5$ weights. In this context, it appears relevant to train a denoiser with only 5 layers ($10^5$ weights), as it is easier to train and less computationally intensive to deploy.

## 5.3 Forward model shift on color images

Theorem 1 has another implication when dealing with uncertainty in the forward model. This arises, for instance, in medical imaging when the exact parameterization of the measuring instrument is not

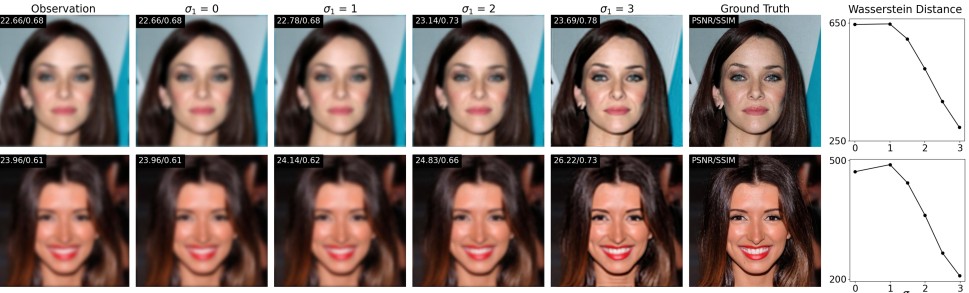

Figure 4: Illustration of the PnP-ULA stability to a mismatch forward model. *Leftmost six plots:* MMSE estimators computed with PnP-ULA run on $30,000$ steps on Gaussian blur of standard deviation $\sigma_1$. *Rightmost:* Evolution of the Wasserstein distance between sampling distributions computed with a mismatched blur kernel, $\sigma_1 \in [0, 3[$ and sampling with the exact forward model $\sigma^\star = 3$. Note that the image reconstruction quality improves as $\sigma_1$ gets closer to $\sigma^\star = 3$, used to degrade the image. In addition, in the case of Gaussian blur, the pseudometric is given by $|\sigma_1 - \sigma^\star|$ which justifies qualitatively the observed linear decrease of the Wasserstein distance.

well-defined (Guerquin-Kern et al., 2012). Corollary 1.3 elaborates on how the sampling distribution is sensitive to shifts in the forward model.

In our experimental setup, we address the deblurring inverse problem using the CelebA validation set, which consists exclusively of images of women's faces resized to RGB dimensions of $256 \times 256$ pixels. A Gaussian blur kernel is applied to the images, with a standard deviation $\sigma_1$ being subject to modification. The denoiser is implemented using the DRUNet architecture (Zhang et al., 2021) and has been trained on the CelebA dataset (Liu et al., 2015). Other parameters of PnP-ULA are chosen following the recommendation of (Laumont et al., 2022).

The image is initially degraded using a Gaussian blur kernel with a standard deviation of $\sigma^\star = 3$. This reference sampling distribution is denoted $\pi^\star$. Multiple Markov chains are then computed, each spanning $30,000$ steps, under the assumption of mismatched forward models with varying standard deviations, $\sigma_1 \in [0, 3[$. At every 30 steps of these chains, we select $N = 1,000$ samples, resulting in distinct sampling distributions $\pi_{\sigma_1}$. We subsequently compute the Wasserstein distance, denoted as $\mathbf{W}_1(\pi^\star, \pi_{\sigma_1})$, to quantify the discrepancy between the reference sampling distribution $\pi^\star$ and the mismatched sampling distribution $\pi_{\sigma_1}$.

This procedure is applied to two images presented in Figure 4. The distance between the blur matrices with a Gaussian kernel is directly the difference between the standard deviation of these kernels. The observed trend aligns precisely with the expected behavior: as the assumed standard deviation of the blur kernel, denoted as $\sigma_1$, approaches 3, the exact kernel of blur, the Wasserstein distance $\mathbf{W}_1(\pi^\star, \pi_{\sigma_1})$ decrease and the quality of the reconstruction improves. It's notable that when $\sigma_1 = 0$, we essentially have a denoising scenario with minimal noise, resulting in a reconstructed image that closely resembles the observed image. This experiment clearly illustrates the corollary 1.3 in context of a forward model mismatch.

## 6 CONCLUSION

In conclusion, our comprehensive analysis of the PnP-ULA algorithm has provided us with insights into the intricacies of this posterior sampling technique. Through rigorous examination, we've successfully unified the understanding of denoiser shifts and variations in the forward model, encapsulating these phenomena in a singular and novel result, denoted as Theorem 1. Importantly, our error bounds are expressed in terms of a posterior-$L_2$ pseudometric, which we show to be more relevant than the previously used bound between denoisers (Laumont et al., 2022). Future work will investigate how to extend our results to other Langevin based dynamics (Klatzer et al., 2023), with a particular focus on annealing and exploring the broader applications of our results in addressing various inverse problems.

# 7 Reproducibility Statement

Source code used in our experiments can be found in PnP ULA posterior law sensivity code. It contains a README.md file that explains step by step how to run the algorithm and replicate the results of the paper. In Section 5 it is precisely detailed how all the hyper-parameters are chosen and, for each experiment, which dataset is used. As for the theoretical results presented in Section 4, complete proofs are given in the Supplements.

# 8 Acknowledgements

This paper is partially based upon work supported by the NSF CAREER award under grants CCF-2043134. This study has been carried out with financial support from the French Research Agency through the PostProdLEAP grant (ANR-19-CE23-0027-01). We also want to thank Shirin Shoushtari for providing pre-trained DRUNet denoisers.

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

SUPPLEMENTARY MATERIAL

Our unified analysis of PnP-ULA is based on stochastic equation theory. In Supplement A, we first analyse the properties of the posterior-$L_2$ pseudometric. In Supplement B, we include additional technical details on experiments. In Supplement C, more result of PnP-ULA on various images are presented. In Supplement D, we demonstrated the Theorem 1. In Supplement E, we derive from the main result the different corollaries.

## A  POSTERIOR-$L_2$ PSEUDOMETRIC

We define the distance which has been naturally introduce 4.1, the posterior-$L_2$ pseudometric :

$$d_1(b_\epsilon^1, b_\epsilon^2) = \sqrt{\mathbb{E}_{X \sim \pi_{\epsilon,\delta}^1} (||b_\epsilon^1(X) - b_\epsilon^2(X)||^2)}.$$

It is clear that this function of $(b_\epsilon^1, b_\epsilon^2)$ is positive, symmetric and is verifies the triangular inequality. However it is not a distance, because the separability is not to be ensured. If $d_1(b_\epsilon^1, b_\epsilon^2) = 0$, then $b_\epsilon^1 = b_\epsilon^2$ on the domain where $\pi_{\epsilon,\delta}^1 > 0$. Thus, $d_1$ will be named posterior-$L_2$ pseudometric.

However, in our case of evaluation it will have the same behavior as a distance. Let's suppose that $\pi_{\epsilon,\delta}^1$ equals $p(\cdot|\boldsymbol{y})$ (no approximation errors). But the support of $p(\cdot|\boldsymbol{y})$ is the support of the prior $p(\boldsymbol{x})$ because of the Gaussian noise. Thus, the two functions are equal if their support are included on the support of the distribution of data. This is totally acceptable because the area of interest is the space of images, i.e. the support of $p(\boldsymbol{x})$.

We use the notation $d_1$ for this metric to emphasize that it involves integrating the $L_2$ norm over the reference sampling distribution $\pi_{\epsilon,\delta}^1$. While it is possible to perform integration over the mismatched sampling distribution $\pi_{\epsilon,\delta}^2$ or even taking the minimum of these two values. We have chosen to retain the expectation only on $\pi_{\epsilon,\delta}^1$ to make it easier to compute in practice. In this work, the convention has been made that $b_\epsilon^1$ is closer to $p_\epsilon(\cdot|\boldsymbol{y})$ than $b_\epsilon^2$ (for instance, with the infinite norm on the space of images). This situation occurs if $D_\epsilon^1$ is trained better than $D_\epsilon^2$ or if the measurement model $\boldsymbol{A}_1$ is more relevant than $\boldsymbol{A}_2$. With this convention, we have observed in practice that the distance computed with $\pi_{\epsilon,\delta}^1$ is more correlated to the Wasserstein distance between the samples than the distance computed with $\pi_{\epsilon,\delta}^2$. This empirical evidence reinforces our choice.

## B  ADDITIONAL TECHNICAL DETAILS

This section presents several technical details that were omitted from the main paper for space. There are three parts for each of the experiments.

### B.1  TECHNICAL DETAILS OF GMM IN 2D

Following the experimental setup of Laumont et al. (2022), a regularization weight $\alpha > 0$ is added to the process. So, the Markov chain, which is computed, is defined by 5 with a drift:

$$b_\epsilon(\boldsymbol{x}) = \nabla \log p(\boldsymbol{y}|\boldsymbol{x}) + \frac{\alpha}{\epsilon} (D_\epsilon(\boldsymbol{x}) - \boldsymbol{x}) + \frac{1}{\lambda} (\Pi_S(\boldsymbol{x}) - \boldsymbol{x}).$$

This parameter allows for the balancing of weights between the prior score and the data-fidelity score. This $\alpha$ parameter is useful in practice to obtain a better convergence. For GMM in 2D, $\alpha = 0.3$. For experiment on images, $\alpha = 1$.

The prior is a Gaussian Mixture Model if

$$p(\boldsymbol{x}) = \sum_{i=1}^{p} w_i \mathcal{N}(\boldsymbol{x}; \boldsymbol{\mu}_i, \boldsymbol{\Sigma}_i),$$

With $w_i \geq 0, \sum_{i=1}^{p} w_i = 1$ weights between the Gaussian and $\mathcal{N}(\boldsymbol{x}; \boldsymbol{\mu}_i, \boldsymbol{\Sigma}_i)$ the Gaussian distribution of mean $\boldsymbol{\mu}_i$ and covariance matrix $\boldsymbol{\Sigma}_i$ evaluate in $\boldsymbol{x}$.

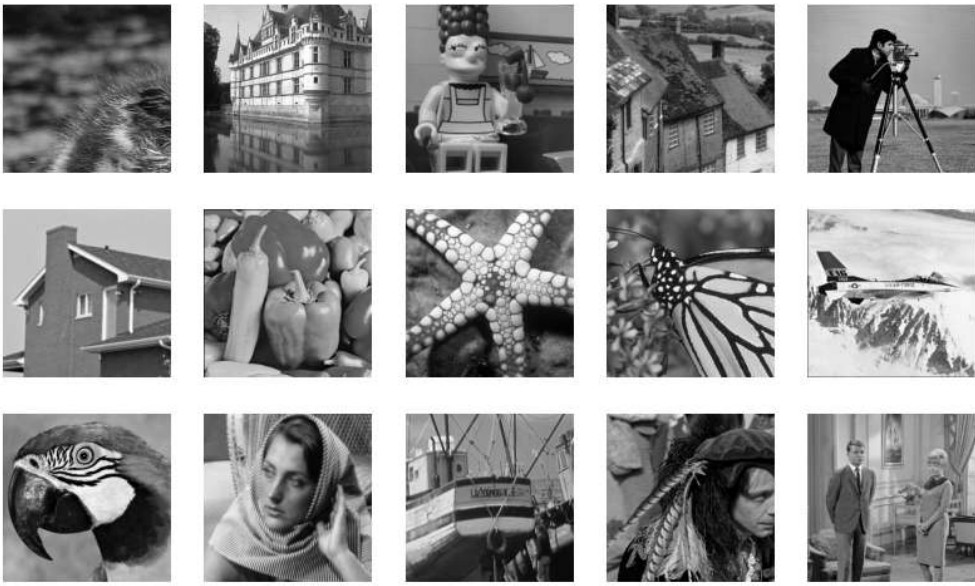

Figure 5: Illustration of the images used in the gray-scale images experiment 5.2

In this case, the posterior distribution has a closed form

$$p(\boldsymbol{x}|\boldsymbol{y}) = \sum_{i=1}^{p} a_i \mathcal{N}(\boldsymbol{x}; \boldsymbol{m}_i, \boldsymbol{S}_i^{-1}),$$

With :

- $\boldsymbol{S}_i = \boldsymbol{\Sigma}_i^{-1} + \frac{\boldsymbol{A}^T \boldsymbol{A}}{\sigma^2}$

- $\boldsymbol{m}_i = \left(\boldsymbol{\Sigma}_i^{-1} + \frac{\boldsymbol{A}^T \boldsymbol{A}}{\sigma^2}\right)^{-1} \left(\boldsymbol{\Sigma}_i^{-1} \boldsymbol{\mu}_i + \frac{\boldsymbol{A}}{\sigma^2} \boldsymbol{y}\right)$

- $a_i = \frac{w_i \exp\left(\frac{1}{2}\boldsymbol{m}_i^T \boldsymbol{S}_i \boldsymbol{m}_i - \frac{1}{2}\boldsymbol{\mu}_i^T \boldsymbol{\Sigma}_i^{-1} \boldsymbol{\mu}_i - \frac{\boldsymbol{y}^T \boldsymbol{y}}{2\sigma^2}\right)}{p(\boldsymbol{y})\sqrt{(2\pi)^n \det\left(\sigma^2 \boldsymbol{I}_n + \boldsymbol{\Sigma}_i^{\frac{1}{2}} \boldsymbol{A}^T \boldsymbol{A} \boldsymbol{\Sigma}_i^{\frac{1}{2}}\right)}}$

Similarly the exact MMSE denoiser $D_\epsilon^\star$ has a closed form

$$D_\epsilon^\star(\boldsymbol{x}) = \frac{\sum_{i=1}^{p} w_i c_i(\boldsymbol{x}) \boldsymbol{n}_i(\boldsymbol{x})}{\sum_{i=1}^{p} w_i c_i(\boldsymbol{x})},$$

With :

- $\boldsymbol{n}_i = \left(\boldsymbol{\Sigma}_i^{-1} + \frac{1}{\epsilon}\boldsymbol{I}_d\right)^{-1} \left(\boldsymbol{\Sigma}_i^{-1} \boldsymbol{\mu}_i + \frac{\boldsymbol{x}}{\epsilon}\right)$

- $c_i = \frac{1}{\sqrt{2\pi \det(\boldsymbol{\Sigma}_i + \epsilon \boldsymbol{I}_d)}} \exp\left(-\frac{1}{2}(\boldsymbol{\mu}_i - \boldsymbol{x})^T (\boldsymbol{\Sigma}_i + \epsilon \boldsymbol{I}_d)^{-1} (\boldsymbol{\mu}_i - \boldsymbol{x})\right)$

In our experiment, parameters of the GMM are $p = 2$, $\boldsymbol{\mu}_1 = \boldsymbol{\mu}_2 = \boldsymbol{0}$, $\boldsymbol{\Sigma}_1 = \begin{pmatrix} 2 & 0.5 \\ 0.5 & 0.15 \end{pmatrix}$, $\boldsymbol{\Sigma}_2 = \begin{pmatrix} 0.15 & 0.5 \\ 0.5 & 2 \end{pmatrix}$, $w_1 = w_2 = 0.5$.

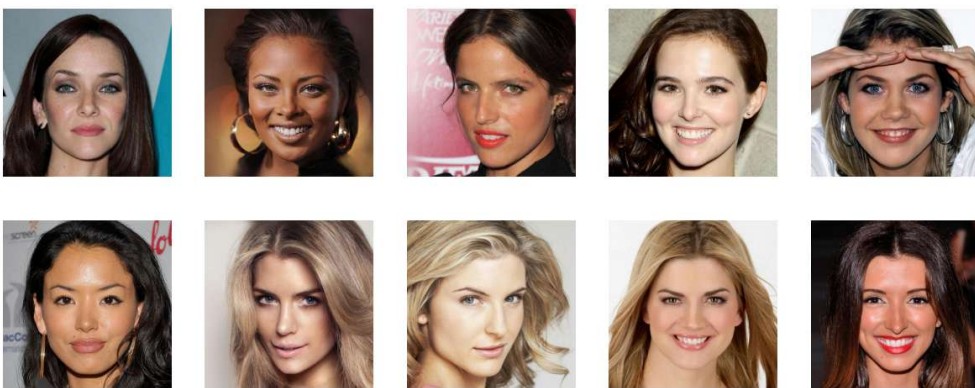

Figure 6: Illustration of images used in the color images experiment 5.3

## B.2  TECHNICAL DETAILS FOR DENOISER SHIFT ON GRAY-SCALE IMAGES

The 15 images used for the gray-scale images experiment 5.2 can be found in the Figure 5. These images are been took from a classical validation set, crop if necessary to be of size $256 * 256$ and normalized within the range of 0 to 1.

A total of 17 distinct DnCNN denoising models (Ryu et al., 2019) were trained, each varying in the number of layers, ranging from a single layer to a maximum of 17 layers. Each of these denoiser is trained on the CBSD68 dataset composed of 400 natural images of size $256 * 256$ for 50 epochs and a learning rate of $1.10^{-3}$. The noise level is fixed at $\frac{5}{255}$ and the Lipschitz constant of the network is not constraint. It has been test to constraint the Lipschitz constant but the network with this constraint perform worst. The computational time required for training one denoiser an NVIDIA GeForce RTX 2080 GPU was approximately 2 hours. So a total of 34 hours of computation was needed to train the gray-scale image denoisers.

The computational time required for one PnP-ULA sampling on an NVIDIA GeForce RTX 2080 GPU was approximately 40 seconds per image. This equates to approximately 3 hours of computational time for the entire experimental procedure. The PnP-ULA is run on in-distribution image (from CBSD68 dataset) and on out-of-distribution images (from BreCaHAD dataset (Aksac et al., 2019) and RxRx1 (Sypetkowski et al., 2023), resized to be 256*256 in gray-scale).

## B.3  TECHNICAL DETAILS FOR DENOISER SHIFT ON COLOR IMAGES

A pretrained denoiser with the DRUNet architecture (Zhang et al., 2021) on CelebA dataset was used. This denoiser was trained with images from CelebA (only women faces) resized to be $256*256$ and with a noise level choose uniformly in the range $[0, \frac{75}{255}]$.

The computational time required for one PnP-ULA sampling of 30,000 steps on an NVIDIA GeForce RTX 2080 GPU was approximately 15 minutes per image. This procedure is applied to a set of 10 images presented in Figure 6. This equates to approximately 15 hours of computational time for the entire experimental procedure. The PnP-ULA is applied on in-distribution images (woman faces from CelebA dataset) and on out-of-distribution dataset (from BreCaHAD dataset, RxRx1 dataset and MetFaces dataset (Karras et al., 2020), resized to be 256*256 color images).

In section 5.3 is only presented result of this experiment on two images but we can compute the mean of the Wasserstein distance $\mathbf{W}_1(\pi_{\sigma_1}, \pi^\star)$ between a sampling with a mismatched measurement model $\sigma_1 \in [0, 3[$ and the exact measurement model $\sigma_1 = 3$. This result is presented in Table B.3.

| Standard deviation of blur kernel | 0 | 1 | 1.5 | 2 | 2.5 |
|---|---|---|---|---|---|
| Wasserstein distance | 640 | 642 | 585 | 471 | 341 |

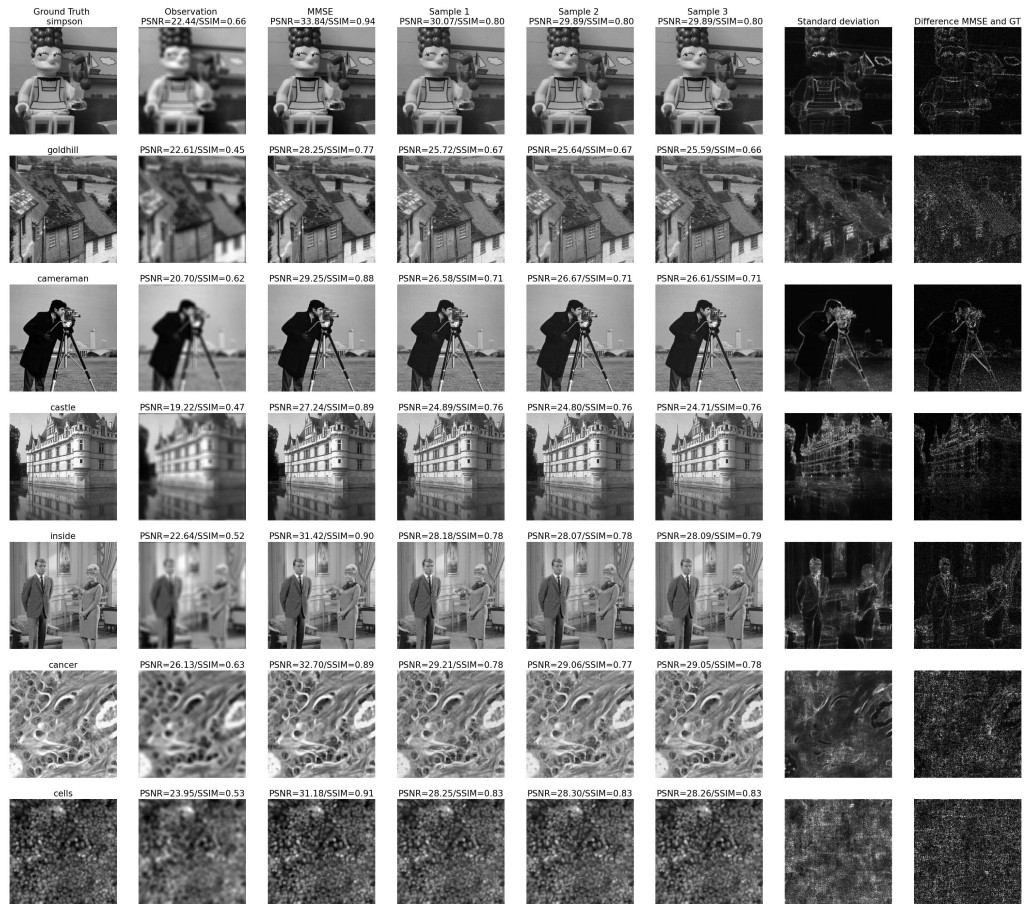

Figure 7: Result of PnP-ULA on different gray scale images for $100,000$ steps with DnCNN. The five top images are in-distribution and the bottom two images are out-of-distribution images.

One can see the same conclusion on this table, the Wasserstein distance decrease when $\sigma_1 \to 3$, showing experimentally the result of Corollary 1.3.

## C  PNP-ULA RESULTS ON IMAGES

In this section, we provide more result of the PnP-ULA sampling algorithm on gray-scale and color images.

**General setting**  Parameters choose for running the PnP-ULA have been taken following the analysis of (Laumont et al., 2022) with $\alpha = 1$, $\epsilon = \frac{5}{255}$, $\lambda = \frac{1}{2(\frac{2}{\sigma^2} + \frac{\alpha}{\epsilon^2})}$ and $\delta = \frac{1}{3(\frac{1}{\sigma^2} + \frac{1}{\lambda} + \frac{\alpha}{\epsilon^2})}$. $C$ has been choose to take a projection on the image space $C = [0,1]^d$ because it gives a little better result in practice than the advice of (Laumont et al., 2022).

The PnP-ULA algorithm is run for deblurring with a uniform blur of $9 * 9$ and a noise level of $\sigma = \frac{1}{255}$. The convergence of the algorithm can be observed in a number of step of order $10,000$ - $100,000$.

**Results on gray-scale images**  The DnCNN denoiser is trained on CBSD68 dataset (natural images of size $256 * 256$ on gray-scale normalized between $0$ and $1$) with $17$ layers and $50$ epochs of training.

In Figure 7, the result on various images can be seen. The image is well reconstruct with the MMSE estimator, especially, looking at the Structural Similarity Index Measure (SSIM), the gap between the observation and the reconstruct image is huge. The algorithm is able to reconstruct more fine structure. By looking at the sample themselves, it is clear that they have a worst quality, this is due to the stochastic term which imply on the sample a residual noise which disappear in the MMSE. By looking at the Standard deviation of the sample, one can remark that it is close to the error map between the MMSE estimator and the ground truth. The outstanding result is that PnP-ULA gives still good result on out-of-distribution data. The MMSE is still relevant but the uncertainty map given by the Standard deviation is not anymore relevant for that kind of data.

One can also remark artefacts on the reconstruction in the roof area of the second image. In fact for this image, if the Markov chain is running for more steps, these artefacts get worse. There is an instability of the algorithm. This might be explain by the fact that the denoiser is not able to constraint the Markov chain for some frequencies who are zeros of the blur kernel (so not constraint also by the data fidelity term). So in these direction in the frequency domain, the Markov chain is free and still discover the space leading artefacts like aliasing.

**Results on color scale**    The DRUNet denoiser is trained on RGB woman faces images of size $256 * 256$ from the dataset celabA. The studied inverse problem and the parameter of the algorithm are the same than for gray scale images. The only modify parameter is the number of step, take at $30,000$, for faster computation.

We have similar result than previously. However the algorithm performance is more visible on the SSIM metric. In this case, the algorithm is also less powerful on out-of-distribution data. Especially on the image of cells, the reconstruction is not good. But the PnP-ULA still gives good results on other out-of-distribution images, which are very different from a woman face. One can remark that the convergence is also a bit weaker. Especially in the different between the MMSE estimator and the ground-truth, artefacts as described before are more visible.

## C.1    Do we capture multimodality with PnP-ULA ?

The problem of capturing the multimodality of the posterior distribution is the key problem of sampling methods (Cohen et al., 2023). Because the inverse problem is ill-posed, multiple proper solutions can correspond to a unique observation. With multiple samples of the posterior law $p(\cdot|\boldsymbol{y})$, we aim to capture the different possibility of reconstruction.

Looking at the eyebrow in Figure 9, we can see that different modes are discovered during the Markov chain process with different number of eyebrows. The posterior distribution seems to be well discovered because this detail is ambiguous in the blurred image implying multimodality of the posterior distribution.

## D    Demonstration of Theorem 1

The proof is based on a decomposition of $\|\pi^1_{\epsilon,\delta} - \pi^2_{\epsilon,\delta}\|_{TV}$ detailed in Equation 11. Then, each term is controlled independently by the exponential convergence of the Markov Chain (Equation 23) and by a consequence of Girsanov's theorem (Equation 15). Finally, the demonstration in Wasserstein distance is deduced from the inequality in the $TV$-norm. Note that the bound in Wasserstein distance might be demonstrated with a similar proof as the one in the $TV$-norm, leading to a more refined inequality.

### D.1    Control of $\|\pi^1_{\epsilon,\delta} - \pi^2_{\epsilon,\delta}\|_{TV}$

**Definition D.1** (The $V$-norm for distributions). The $V$-norm (for $V : \mathbb{R}^d \mapsto [1, +\infty[$) of a distribution is defined by

$$\|\pi\|_V = \sup_{\|f/V\|_\infty \leq 1} \left| \int_{\mathbb{R}^d} f(\boldsymbol{x}) d\pi(\boldsymbol{x}) \right|.$$

The $TV$-norm is the $V$-norm with $V = 1$. For every distribution $\pi$ on $\mathbb{R}^d$, $\|\pi\|_{TV} \leq \|\pi\|_V$.

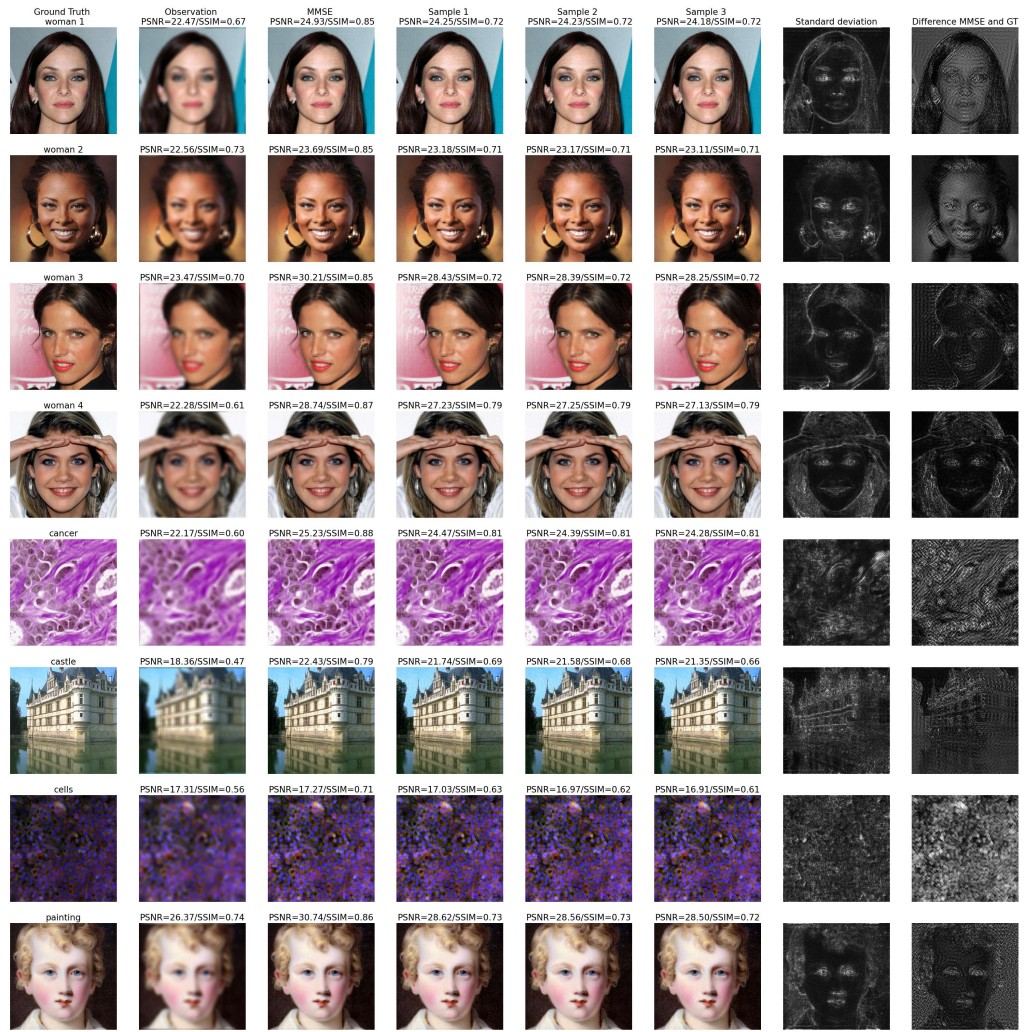

Figure 8: Result of PnP-ULA on different color images for $30,000$ steps with Drunet. Top four images are in-distribution and bottom fouor images are out-of-distribution images.

### D.1.1 FIRST DECOMPOSITION

The Markov kernel of the Markov chain is defined by, for $i \in \{1, 2\}$ :

$$R_{\epsilon,\delta}^i(\boldsymbol{x}, \mathbb{A}) = \frac{1}{(2\pi)^{\frac{d}{2}}} \int_{\mathbb{R}^d} \mathbb{1}_{\mathbb{A}} \left( \boldsymbol{x} + \delta b_\epsilon^i(\boldsymbol{x}) + \sqrt{2\delta} \boldsymbol{z} \right) \exp \left( -\frac{\|\boldsymbol{z}\|^2}{2} \right) d\boldsymbol{z}.$$

$R_{\epsilon,\delta}^i(\boldsymbol{x}, \mathbb{A})$ is the probability that the Markov Chain in $\boldsymbol{x}$ go in the set $\mathbb{A}$ at next iteration. The Markov Kernel is the transition matrix of the Markov Chain if the number of state is finite. Here the space of state is $\mathbb{R}^d$, leading to this continuous definition.

By definition of the invariant law, because the Markov chain $(\boldsymbol{x}_k^i)_{k \in \mathbb{N}}$ converges in law to $\pi_{\epsilon,\delta}^i$, we gets $\forall N \in \mathbb{N}, \pi_{\epsilon,\delta}^i \left( R_{\epsilon,\delta}^i \right)^N = \pi_{\epsilon,\delta}^i$.

The first step of the bounding of $\|\pi_{\epsilon,\delta}^1 - \pi_{\epsilon,\delta}^2\|_{TV}$ is to remark that, $\forall N \in \mathbb{N}, \forall \boldsymbol{x} \in \mathbb{R}^d$ :

$$\|\pi_{\epsilon,\delta}^1 - \pi_{\epsilon,\delta}^2\|_{TV} \le \|\delta_{\boldsymbol{x}}(R_{\epsilon,\delta}^1)^N - \pi_{\epsilon,\delta}^1\|_{TV} + \|\delta_{\boldsymbol{x}}(R_{\epsilon,\delta}^1)^N - \delta_{\boldsymbol{x}}(R_{\epsilon,\delta}^2)^N\|_{TV} + \|\delta_{\boldsymbol{x}}(R_{\epsilon,\delta}^2)^N - \pi_{\epsilon,\delta}^2\|_{TV}.$$
$$(11)$$

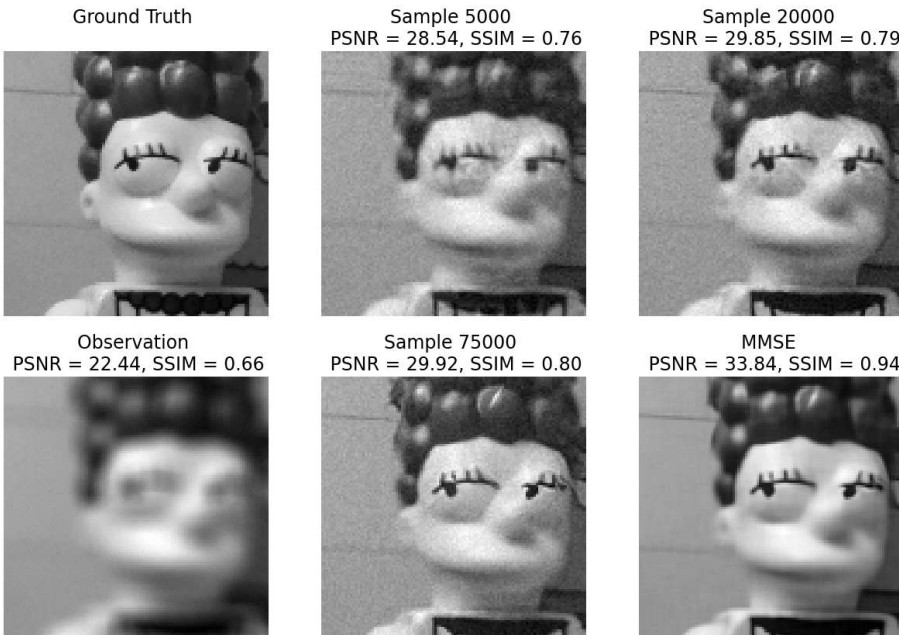

Figure 9: Patches of different sample from one PnP-ULA Markov chain run on image Simpson

$\delta_{\boldsymbol{x}}(R^1_{\epsilon,\delta})^N$ is the probability density of the $N$ iterate of the Markov chain which begin in $\boldsymbol{x}$ (so with the distribution $\delta_{\boldsymbol{x}}$).

We will first bound separately $\|\delta_{\boldsymbol{x}}(R^i_{\epsilon,\delta})^N - \pi^i_{\epsilon,\delta}\|_{TV}$ and $\|\delta_{\boldsymbol{x}}(R^1_{\epsilon,\delta})^N - \delta_{\boldsymbol{x}}(R^2_{\epsilon,\delta})^N\|_{TV}$, then deduce a bound on the target quantity.

### D.1.2 BOUND OF $\|\delta_{\boldsymbol{x}}(R^i_{\epsilon,\delta})^N - \pi^i_{\epsilon,\delta}\|_{TV}$

By (Laumont et al., 2022, Proposition 5), $\forall \boldsymbol{x} \in \mathbb{R}^d$, there exists $a_1, a_2 \in \mathbb{R}^+$ and $\rho_1, \rho_2 \in ]0,1[$, such that, for $i \in \{1,2\}$ :

$$\|\delta_{\boldsymbol{x}}(R^i_{\epsilon,\delta})^N - \pi^i_{\epsilon,\delta}\|_{TV} \leq \|\delta_{\boldsymbol{x}}(R^i_{\epsilon,\delta})^N - \pi^i_{\epsilon,\delta}\|_V \leq a_i \rho_i^{N\delta} \left( V^2(\boldsymbol{x}) + \int_{\mathbb{R}^d} V^2(\tilde{\boldsymbol{x}}) \pi^i_{\epsilon,\delta}(d\tilde{\boldsymbol{x}}) \right),$$

with the $V$-norm defined in Definition D.1.

### D.1.3 BOUND OF $\|\delta_{\boldsymbol{x}}(R^1_{\epsilon,\delta})^N - \delta_{\boldsymbol{x}}(R^2_{\epsilon,\delta})^N\|_{TV}$

We will suppose that $\delta = \frac{1}{m}$ and control the quantity defined for $k \in \mathbb{N}$ by

$$\|\delta_{\boldsymbol{x}}(R^1_{\epsilon,\delta})^{km} - \delta_{\boldsymbol{x}}(R^2_{\epsilon,\delta})^{km}\|_{TV}.$$

We can make the decomposition

$$\|\delta_{\boldsymbol{x}}(R^1_{\epsilon,\frac{1}{m}})^{km} - \delta_{\boldsymbol{x}}(R^2_{\epsilon,\frac{1}{m}})^{km}\|_{TV} = \|\sum_{j=0}^{k-1} \delta_{\boldsymbol{x}}(R^1_{\epsilon,\frac{1}{m}})^{(k-j-1)m} \left( (R^1_{\epsilon,\frac{1}{m}})^m - (R^2_{\epsilon,\frac{1}{m}})^m \right) (R^2_{\epsilon,\frac{1}{m}})^{jm}\|_{TV}. \tag{12}$$

We introduce the continuous Markov processes, $(\tilde{\boldsymbol{x}}^i_t)_{t \in [0,m\delta]}$ for $i \in \{1,2\}$, defined to be the strong solution of :

$$d\tilde{\boldsymbol{x}}^i_t = \tilde{b}^i_\epsilon(t, (\tilde{\boldsymbol{x}}^i_t)_{t \in [0,m\delta]}) + \sqrt{2}d\tilde{\boldsymbol{b}}^i_t, \tag{13}$$

With $\tilde{\boldsymbol{x}}_0^i \sim \delta_{\boldsymbol{x}}$, $\tilde{b}_\epsilon^i(t, (\tilde{\boldsymbol{x}}_t^i)_{t\in[0,m\delta]}) = \sum_{j=0}^{m-1} \mathbb{1}_{[j\delta,(j+1)\delta[}(t) b_\epsilon^i(\tilde{\boldsymbol{x}}_{j\delta}^i)$ and $\tilde{b}_t^i$ a continuous Wiener process. For simplicity of notation, in the following computation we will write $\tilde{b}_\epsilon^i(\tilde{\boldsymbol{x}}_t^i)$ instead of $\tilde{b}_\epsilon^i(t, (\tilde{\boldsymbol{x}}_t^i)_{t\in[0,m\delta]})$.

From this definition, $\tilde{\boldsymbol{x}}_{k\delta}^i = \boldsymbol{x}_k^i$ for $k \in [0, m-1]$. And one can remark, that $m\delta = 1$.

**Moment control**  First we control the moment of $\tilde{\boldsymbol{x}}_t^i$. The moment bound, $\forall j \in \mathbb{N}^*$, with for $i \in \{0, 1\}$ is defined by

$$m_j = \max \left( \sup_{t\in[0,+\infty[} \left( \mathbb{E}_{\mathbf{x}\sim\delta_{\boldsymbol{x}}P_t^1}[\|\mathbf{x}\|^j] \right), \sup_{t\in[0,+\infty[} \left( \mathbb{E}_{\mathbf{y}\sim\delta_{\boldsymbol{x}}P_t^2}[\|\mathbf{y}\|^j] \right) \right),$$

with $(P_t^i)_{t\in[0,+\infty[}$ the Markov kernels associated with process 13.

For $W_j(\boldsymbol{x}) = 1 + \|x\|^j$. By (Laumont et al., 2022, Lemma 17), with $\bar{\delta} = \frac{1}{3}\left(L + \frac{M+1}{\epsilon} + \frac{1}{\lambda}\right)^{-1}$, $\exists \lambda \in [0, 1[$, such that $\exists C_j \geq 0$, such that $\forall i \in \{1, 2\}$, $\forall \delta \in ]0, \bar{\delta}]$, $\forall \boldsymbol{x} \in \mathbb{R}^d$, $\forall k \in \mathbb{N}^*$:

$$P_t^i W_j \leq C_j W_j$$
$$\delta_{\boldsymbol{x}} P_t^i W_j \leq C_i \delta_{\boldsymbol{x}} W_j$$
$$1 + \mathbb{E}_{\mathbf{x}\sim\delta_{\boldsymbol{x}}P_t^1}[\|\mathbf{x}\|^j] \leq C_j W_j(\boldsymbol{x})$$
$$\mathbb{E}_{\mathbf{x}\sim\delta_{\boldsymbol{x}}P_t^i}[\|\mathbf{x}\|^j] \leq C_j W_j(\boldsymbol{x})$$
$$\sup_{t\in[0,+\infty[} \left( \mathbb{E}_{\mathbf{x}\sim\delta_{\boldsymbol{x}}P_t^i}[\|\mathbf{x}\|^j] \right) \leq C_j W_j(\boldsymbol{x}).$$

Note that $C_j W_j(\boldsymbol{x})$ is independent of $\delta$. It has been shown that : $\exists M_j \geq 0$ independent of $\delta$ such as $\forall \delta \in ]0, \bar{\delta}]$ :

$$m_j \leq M_j W_j(\boldsymbol{x}). \tag{14}$$

**Control of** $\|\delta_{\boldsymbol{x}}(R_{\epsilon,\frac{1}{m}}^1)^m - \delta_{\boldsymbol{x}}(R_{\epsilon,\frac{1}{m}}^2)^m\|_V$

For any $i \in \{1, 2\}$, we have

$$\|\delta_{\boldsymbol{x}}(R_{\epsilon,\frac{1}{m}}^1)^m - \delta_{\boldsymbol{x}}(R_{\epsilon,\frac{1}{m}}^2)^m\|_V = \|\delta_{\boldsymbol{x}}P_1^1 - \delta_{\boldsymbol{x}}P_1^2\|_V,$$

with $(P_t^i)_{t\in[0,+\infty[}$ the Markov kernels associated with process 13.

Using (Laumont et al., 2022, Lemma 19), which is a direct consequence of the Girsanov's theorem (Shiryayev, 1977, Theorem 7.7), we get the bound

$$\|\delta_{\boldsymbol{x}}(R_{\epsilon,\frac{1}{m}}^1)^m - \delta_{\boldsymbol{x}}(R_{\epsilon,\frac{1}{m}}^2)^m\|_V \leq \sqrt{2} \left( \int_0^1 \mathbb{E}\left( \|\tilde{b}_\epsilon^1(\tilde{\boldsymbol{x}}_t^1) - \tilde{b}_\epsilon^2(\tilde{\boldsymbol{x}}_t^1)\|^2 \right) dt \right)^{\frac{1}{2}}. \tag{15}$$

**Estimation of** $\sqrt{\int_0^1 \mathbb{E}\left( \|\tilde{b}_\epsilon^1(\tilde{\boldsymbol{x}}_t^1) - \tilde{b}_\epsilon^2(\tilde{\boldsymbol{x}}_t^1)\|^2 \right) dt}$

For $t \in [0, 1]$, thanks to the form of $\tilde{b}_\epsilon$, $\tilde{\boldsymbol{x}}_{\lfloor\frac{t}{\delta}\rfloor\delta}$ has the same law that $\boldsymbol{x}_{\lfloor\frac{t}{\delta}\rfloor}$.

we can deduce from the previous property an estimation of the error

$$\left| \mathbb{E}\left( \|\tilde{b}_\epsilon^1(\tilde{\boldsymbol{x}}_t^1) - \tilde{b}_\epsilon^2(\tilde{\boldsymbol{x}}_t^1)\|^2 \right) - \mathbb{E}\left( \|b_\epsilon^1(\tilde{\boldsymbol{x}}_{\lfloor\frac{t}{\delta}\rfloor\delta}^1) - b_\epsilon^2(\tilde{\boldsymbol{x}}_{\lfloor\frac{t}{\delta}\rfloor\delta}^1)\|^2 \right) \right| \tag{16}$$

$$= \left| \mathbb{E}\left( \|\tilde{b}_\epsilon^1(\tilde{\boldsymbol{x}}_t^1) - \tilde{b}_\epsilon^2(\tilde{\boldsymbol{x}}_t^1)\| \right) - \mathbb{E}\left( \|b_\epsilon^1(\tilde{\boldsymbol{x}}_{\lfloor\frac{t}{\delta}\rfloor\delta}^1) - b_\epsilon^2(\tilde{\boldsymbol{x}}_{\lfloor\frac{t}{\delta}\rfloor\delta}^1)\| \right) \right| \left| \mathbb{E}\left( \|\tilde{b}_\epsilon^1(\tilde{\boldsymbol{x}}_t^1) - \tilde{b}_\epsilon^2(\tilde{\boldsymbol{x}}_t^1)\| \right) + \mathbb{E}\left( \|b_\epsilon^1(\tilde{\boldsymbol{x}}_{\lfloor\frac{t}{\delta}\rfloor\delta}^1) - b_\epsilon^2(\tilde{\boldsymbol{x}}_{\lfloor\frac{t}{\delta}\rfloor\delta}^1)\| \right) \right|. \tag{17}$$

The first term can be control by the fact that $b_\epsilon^1$ and $b_\epsilon^2$ are $L_b = \frac{M+1}{\epsilon} + L + \frac{1}{\lambda}$-Lipschitz, which immediately follows from the Assumptions 1 and 3

$$
\begin{aligned}
&\left| \mathbb{E}\left( \|\tilde{b}_\epsilon^1(\tilde{\boldsymbol{x}}_t^1) - \tilde{b}_\epsilon^2(\tilde{\boldsymbol{x}}_t^1)\| \right) - \mathbb{E}\left( \|b_\epsilon^1(\tilde{\boldsymbol{x}}_{\lfloor \frac{t}{\delta} \rfloor \delta}^1) - b_\epsilon^2(\tilde{\boldsymbol{x}}_{\lfloor \frac{t}{\delta} \rfloor \delta}^1)\| \right) \right| \\
&\leq \mathbb{E}\left( \left| \|\tilde{b}_\epsilon^1(\tilde{\boldsymbol{x}}_t^1) - \tilde{b}_\epsilon^2(\tilde{\boldsymbol{x}}_t^1)\| - \|b_\epsilon^1(\tilde{\boldsymbol{x}}_{\lfloor \frac{t}{\delta} \rfloor \delta}^1) - b_\epsilon^2(\tilde{\boldsymbol{x}}_{\lfloor \frac{t}{\delta} \rfloor \delta}^1)\| \right| \right) \\
&\leq \mathbb{E}\left( \left| \|\tilde{b}_\epsilon^1(\tilde{\boldsymbol{x}}_t^1) - \tilde{b}_\epsilon^2(\tilde{\boldsymbol{x}}_t^1)\| - \|b_\epsilon^1(\tilde{\boldsymbol{x}}_{\lfloor \frac{t}{\delta} \rfloor \delta}^1) - b_\epsilon^2(\tilde{\boldsymbol{x}}_{\lfloor \frac{t}{\delta} \rfloor \delta}^1)\| \right| \right) \\
&\leq \mathbb{E}\left( \|\tilde{b}_\epsilon^1(\tilde{\boldsymbol{x}}_t^1) - b_\epsilon^1(\tilde{\boldsymbol{x}}_{\lfloor \frac{t}{\delta} \rfloor \delta}^1) + b_\epsilon^2(\tilde{\boldsymbol{x}}_{\lfloor \frac{t}{\delta} \rfloor \delta}^1) - \tilde{b}_\epsilon^2(\tilde{\boldsymbol{x}}_t^1)\| \right) \\
&\leq 2L_b \mathbb{E}\left( \|\tilde{\boldsymbol{x}}_t^1 - \tilde{\boldsymbol{x}}_{\lfloor \frac{t}{\delta} \rfloor \delta}^1\| \right).
\end{aligned}
$$

$b_\epsilon^i$ is Lipschitz, so there exists $D_\epsilon \leq 0$, such that $\forall \boldsymbol{x} \in \mathbb{R}^d$, $\forall i \in \{1, 2\}$,

$$
b_\epsilon^i(\boldsymbol{x}) \leq D_\epsilon(1 + \|\boldsymbol{x}\|). \tag{18}
$$

By the Itô's isometry, Equation 18 and Equation 14, we get

$$
\begin{aligned}
\mathbb{E}\left( \|\tilde{\boldsymbol{x}}_t^1 - \tilde{\boldsymbol{x}}_{\lfloor \frac{t}{\delta} \rfloor \delta}^1\|^2 \right) &\leq \mathbb{E}\left( \| \int_{\tilde{\boldsymbol{x}}_{\lfloor \frac{t}{\delta} \rfloor \delta}^1}^{\tilde{\boldsymbol{x}}_t^1} \tilde{b}_\epsilon^1(u) du \|^2 \right) + 2\mathbb{E}\left( \| \int_{\tilde{\boldsymbol{x}}_{\lfloor \frac{t}{\delta} \rfloor \delta}^1}^{\tilde{\boldsymbol{x}}_t^1} d\tilde{\boldsymbol{b}}_t^1 \|^2 \right) \\
&\leq (2d + 2D_\epsilon(1 + M_2 W_2(\boldsymbol{x})))\delta.
\end{aligned}
$$

The previous inequality gives that

$$
\left| \mathbb{E}\left( \|\tilde{b}_\epsilon^1(\tilde{\boldsymbol{x}}_t^1) - \tilde{b}_\epsilon^2(\tilde{\boldsymbol{x}}_t^1)\| \right) - \mathbb{E}\left( \|b_\epsilon^1(\tilde{\boldsymbol{x}}_{\lfloor \frac{t}{\delta} \rfloor \delta}^1) - b_\epsilon^2(\tilde{\boldsymbol{x}}_{\lfloor \frac{t}{\delta} \rfloor \delta}^1)\| \right) \right| \leq 2L_b \sqrt{(2d + 2D_\epsilon(1 + M_2 W_2(\boldsymbol{x})))\delta}. \tag{19}
$$

Now we control the second term of the error estimation

$$
\begin{aligned}
&\left| \mathbb{E}\left( \|\tilde{b}_\epsilon^1(\tilde{\boldsymbol{x}}_t^1) - \tilde{b}_\epsilon^2(\tilde{\boldsymbol{x}}_t^1)\| \right) + \mathbb{E}\left( \|b_\epsilon^1(\tilde{\boldsymbol{x}}_{\lfloor \frac{t}{\delta} \rfloor \delta}^1) - b_\epsilon^2(\tilde{\boldsymbol{x}}_{\lfloor \frac{t}{\delta} \rfloor \delta}^1)\| \right) \right| \\
&\leq \mathbb{E}\left( \|\tilde{b}_\epsilon^1(\tilde{\boldsymbol{x}}_t^1)\| \right) + \mathbb{E}\left( \|\tilde{b}_\epsilon^2(\tilde{\boldsymbol{x}}_t^1)\| \right) + \mathbb{E}\left( \|b_\epsilon^1(\tilde{\boldsymbol{x}}_{\lfloor \frac{t}{\delta} \rfloor \delta}^1)\| \right) + \mathbb{E}\left( \|b_\epsilon^2(\tilde{\boldsymbol{x}}_{\lfloor \frac{t}{\delta} \rfloor \delta}^1)\| \right) \\
&\leq 2D_\epsilon \left( 2 + \|\tilde{\boldsymbol{x}}_t^1\| + \|\tilde{\boldsymbol{x}}_{\lfloor \frac{t}{\delta} \rfloor \delta}^1\| \right) \\
&\leq 4D_\epsilon \left( 1 + M_1 W_1(\boldsymbol{x}) \right).
\end{aligned}
$$

Thus the second part of the error estimation is bounded by

$$
\left| \mathbb{E}\left( \|\tilde{b}_\epsilon^1(\tilde{\boldsymbol{x}}_t^1) - \tilde{b}_\epsilon^2(\tilde{\boldsymbol{x}}_t^1)\| \right) + \mathbb{E}\left( \|b_\epsilon^1(\tilde{\boldsymbol{x}}_{\lfloor \frac{t}{\delta} \rfloor \delta}^1) - b_\epsilon^2(\tilde{\boldsymbol{x}}_{\lfloor \frac{t}{\delta} \rfloor \delta}^1)\| \right) \right| \leq 4D_\epsilon \left( 1 + M_1 W_1(\boldsymbol{x}) \right). \tag{20}
$$

Using Equation 16, Equation 19 and Equation 20 we obtain the error estimation

$$
\left| \mathbb{E}\left( \|\tilde{b}_\epsilon^1(\tilde{\boldsymbol{x}}_t^1) - \tilde{b}_\epsilon^2(\tilde{\boldsymbol{x}}_t^1)\|^2 \right) - \mathbb{E}\left( \|b_\epsilon^1(\tilde{\boldsymbol{x}}_{\lfloor \frac{t}{\delta} \rfloor \delta}^1) - b_\epsilon^2(\tilde{\boldsymbol{x}}_{\lfloor \frac{t}{\delta} \rfloor \delta}^1)\|^2 \right) \right| \leq 8D_\epsilon L_b G(\boldsymbol{x})\sqrt{\delta},
$$

With $G(\boldsymbol{x}) = \sqrt{(2d + 2D_\epsilon(1 + M_2 W_2(\boldsymbol{x})))}(1 + M_1 W_1(\boldsymbol{x}))$.

Hence, we get an estimation of the integral in Equation 15

$$
\left| \sqrt{\int_0^1 \mathbb{E}\left( \|\tilde{b}_\epsilon^1(\tilde{\boldsymbol{x}}_t^1) - \tilde{b}_\epsilon^2(\tilde{\boldsymbol{x}}_t^1)\|^2 \right) dt} - \sqrt{\int_0^1 \mathbb{E}\left( \|b_\epsilon^1(\tilde{\boldsymbol{x}}_{\lfloor \frac{t}{\delta} \rfloor \delta}^1) - b_\epsilon^2(\tilde{\boldsymbol{x}}_{\lfloor \frac{t}{\delta} \rfloor \delta}^1)\|^2 \right) dt} \right| \leq \sqrt{8D_\epsilon L_b G(\boldsymbol{x})} \delta^{\frac{1}{4}}. \tag{21}
$$

**Control of $\|\delta_{\boldsymbol{x}}(R^1_{\epsilon,\frac{1}{m}})^m - \delta_{\boldsymbol{x}}(R^2_{\epsilon,\frac{1}{m}})^m\|_V$**

Now we will deduce from the previous paragraph a bound on $\|\delta_{\boldsymbol{x}}(R^1_{\epsilon,\frac{1}{m}})^m - \delta_{\boldsymbol{x}}(R^2_{\epsilon,\frac{1}{m}})^m\|_{TV}$. Using Equation 21 and that $\tilde{\boldsymbol{x}}_{\lfloor \frac{t}{\delta} \rfloor \delta}$ has the same law that $\boldsymbol{x}_{\lfloor \frac{t}{\delta} \rfloor}$

$$\|\delta_{\boldsymbol{x}}(R^1_{\epsilon,\frac{1}{m}})^m - \delta_{\boldsymbol{x}}(R^2_{\epsilon,\frac{1}{m}})^m\|_V \le \sqrt{2}\sqrt{\int_0^1 \mathbb{E}\left(\|b^1_\epsilon(\tilde{\boldsymbol{x}}^1_{\lfloor \frac{t}{\delta} \rfloor \delta}) - b^2_\epsilon(\tilde{\boldsymbol{x}}^1_{\lfloor \frac{t}{\delta} \rfloor \delta})\|^2\right) dt} + \sqrt{16 D_\epsilon L_b G(\boldsymbol{x})}\delta^{\frac{1}{4}}$$

$$\le \sqrt{2}\sqrt{\int_0^1 \mathbb{E}\left(\|b^1_\epsilon(\boldsymbol{x}^1_{\lfloor \frac{t}{\delta} \rfloor}) - b^2_\epsilon(\boldsymbol{x}^1_{\lfloor \frac{t}{\delta} \rfloor})\|^2\right) dt} + \sqrt{16 D_\epsilon L_b G(\boldsymbol{x})}\delta^{\frac{1}{4}}$$

$$\le \sqrt{2}\sqrt{\frac{1}{m}\sum_{l=0}^{m-1} \mathbb{E}\left(\|b^1_\epsilon(\boldsymbol{x}^1_l) - b^2_\epsilon(\boldsymbol{x}^1_l)\|^2\right)} + \sqrt{16 D_\epsilon L_b G(\boldsymbol{x})}\delta^{\frac{1}{4}}.$$

Hence, we denote $D_0 = \sqrt{16 D_\epsilon L_b}$ and it gives

$$\|\delta_{\boldsymbol{x}}(R^1_{\epsilon,\frac{1}{m}})^m - \delta_{\boldsymbol{x}}(R^2_{\epsilon,\frac{1}{m}})^m\|_V \le \sqrt{2}\sqrt{\frac{1}{m}\sum_{l=0}^{m-1} \mathbb{E}\left(\|b^1_\epsilon(\boldsymbol{x}^1_l) - b^2_\epsilon(\boldsymbol{x}^1_l)\|^2\right)} + D_0\sqrt{G(\boldsymbol{x})}\delta^{\frac{1}{4}}. \quad (22)$$

**Control of $\|\delta_{\boldsymbol{x}}(R^1_{\epsilon,\frac{1}{m}})^{km} - \delta_{\boldsymbol{x}}(R^2_{\epsilon,\frac{1}{m}})^{km}\|_V$**

Using Laumont et al. (2022, Proposition 10), there exist $A_C \ge 0$ and $\rho_C \in [0, 1[$ such that for any $\boldsymbol{x}, \boldsymbol{y} \in \mathbb{R}^d$, $N \in \mathbb{N}$ and $i \in \{1, 2\}$:
$$\|\delta_{\boldsymbol{x}}(R^i_{\epsilon,\frac{1}{m}})^N - \delta_{\boldsymbol{y}}(R^i_{\epsilon,\frac{1}{m}})^N\|_V \le A_C \rho_C^N \left(V^2(\boldsymbol{x}) + V^2(\boldsymbol{y})\right).$$

For $f : \mathbb{R}^d \mapsto \mathbb{R}$ measurable such as $\forall \boldsymbol{x} \in \mathbb{R}^d$, $|f(\boldsymbol{x})| \le V(\boldsymbol{x})$. This result combine with (Laumont et al., 2022, Lemma 17), shows that there exists $B_a \le 0$ such that $\forall x \in \mathbb{R}^d$, $\forall N \in \mathbb{N}$, $\forall m \in \mathbb{N}$ and $\forall i \in \{1, 2\}$:
$$\left|\delta_{\boldsymbol{x}}(R^i_{\epsilon,\frac{1}{m}})^N[f] - \pi^i_{\epsilon,\frac{1}{m}}[f]\right| \le B_a \rho_C^N V^2(\boldsymbol{x}). \quad (23)$$

Using Equation 22 and Equation 23, we have :
$$\left|\delta_{\boldsymbol{x}}\left((R^1_{\epsilon,\frac{1}{m}})^m - (R^2_{\epsilon,\frac{1}{m}})^m\right)(R^2_{\epsilon,\frac{1}{m}})^{jm}[f]\right| = \left|\delta_{\boldsymbol{x}}\left((R^1_{\epsilon,\frac{1}{m}})^m - (R^2_{\epsilon,\frac{1}{m}})^m\right)\left((R^2_{\epsilon,\frac{1}{m}})^{jm}[f] - \pi^i_{\epsilon,\frac{1}{m}}[f]\right)\right|$$

$$= B_a \rho_C^{jm} V(\boldsymbol{x})\left|\delta_{\boldsymbol{x}}\left((R^1_{\epsilon,\frac{1}{m}})^m - (R^2_{\epsilon,\frac{1}{m}})^m\right)\left(\frac{(R^2_{\epsilon,\frac{1}{m}})^{jm}[f] - \pi^i_{\epsilon,\frac{1}{m}}[f]}{B_a \rho_C^{jm} V(\boldsymbol{x})}\right)\right|$$

$$\le B_a \rho_C^{jm} V(\boldsymbol{x})\|\delta_{\boldsymbol{x}}\left((R^1_{\epsilon,\frac{1}{m}})^m - (R^2_{\epsilon,\frac{1}{m}})^m\right)\|_V$$

$$\le \sqrt{2} B_a \rho_C^{jm} V(\boldsymbol{x})\sqrt{\frac{1}{m}\sum_{l=0}^{m-1} \mathbb{E}\left(\|b^1_\epsilon(\boldsymbol{x}^1_l) - b^2_\epsilon(\boldsymbol{x}^1_l)\|^2\right)} + D_0\sqrt{G(\boldsymbol{x})} B_a \rho_C^{jm} V(\boldsymbol{x})\delta^{\frac{1}{4}}.$$

In the last inequality, the Markov chain $\boldsymbol{x}^1_l$ has be initialized with the distribution $\delta_{\boldsymbol{x}}$. In the following, we will denote $\boldsymbol{x}^1_{l,k}$ the Markov chain defined in 13 with the initial distribution $\delta_{\boldsymbol{x}}(R^1_{\epsilon,\frac{1}{m}})^k$. By using the decomposition in Equation 12 and the previous computation, we get

$$\|\delta_{\boldsymbol{x}}(R^1_{\epsilon,\frac{1}{m}})^{km} - \delta_{\boldsymbol{x}}(R^2_{\epsilon,\frac{1}{m}})^{km}\|_V \le \sum_{j=0}^{k-1} \|\delta_{\boldsymbol{x}}(R^1_{\epsilon,\frac{1}{m}})^{(k-j-1)m}\left((R^1_{\epsilon,\frac{1}{m}})^m - (R^2_{\epsilon,\frac{1}{m}})^m\right)(R^2_{\epsilon,\frac{1}{m}})^{jm}\|_V$$

$$\le \sum_{j=0}^{k-1} \sqrt{2} B_a \rho_C^{jm} V(\boldsymbol{x})\sqrt{\frac{1}{m}\sum_{l=0}^{m-1} \mathbb{E}\left(\|b^1_\epsilon(\boldsymbol{x}^1_{l,(k-j-1)m}) - b^2_\epsilon(\boldsymbol{x}^1_{l,(k-j-1)m})\|^2\right)}$$

$$+ D_0 B_a \rho_C^{jm} \delta^{\frac{1}{4}} V(\boldsymbol{x}) \mathbb{E}_{\mathbf{x} \sim \delta_{\boldsymbol{x}}(R^1_{\epsilon,\frac{1}{m}})^{(k-j-1)m}}\left(\sqrt{G(\mathbf{x})}\right).$$

By the Jensen inequality, the inequality $\mathbb{E}(X) \leq \sqrt{\mathbb{E}(X^2)}$ and the moment control (Equation 14), there exists $g(\boldsymbol{x}) < +\infty$ such that :

$$\mathbb{E}_{\mathbf{x} \sim \delta_{\boldsymbol{x}}(R^1_{\epsilon,\frac{1}{m}})^{(k-j-1)m}} \left( \sqrt{G(\mathbf{x})} \right) = \mathbb{E}_{\mathbf{x} \sim \delta_{\boldsymbol{x}}(R^1_{\epsilon,\frac{1}{m}})^{(k-j-1)m}} \left( \sqrt{\sqrt{(2d + 2D_\epsilon(1 + M_2 W_2(\mathbf{x})))}(1 + M_1 W_1(\mathbf{x}))} \right)$$

$$\leq \sqrt{\mathbb{E}_{\mathbf{x} \sim \delta_{\boldsymbol{x}}(R^1_{\epsilon,\frac{1}{m}})^{(k-j-1)m}} \left( \sqrt{(2d + 2D_\epsilon(1 + M_2 W_2(\mathbf{x})))}(1 + M_1 W_1(\mathbf{x})) \right)}$$

$$\leq \left( \mathbb{E}_{\mathbf{x} \sim \delta_{\boldsymbol{x}}(R^1_{\epsilon,\frac{1}{m}})^{(k-j-1)m}} \left( (2d + 2D_\epsilon(1 + M_2 W_2(\mathbf{x})))(1 + M_1 W_1(\mathbf{x}))^2 \right) \right)^{\frac{1}{4}}$$

$$\leq \left( \mathbb{E}_{\mathbf{x} \sim \delta_{\boldsymbol{x}}(R^1_{\epsilon,\frac{1}{m}})^{(k-j-1)m}} \left( 2(2d + 2D_\epsilon(1 + M_2 W_2(\mathbf{x})))(1 + M_1^2 W_1(\mathbf{x})^2) \right) \right)^{\frac{1}{4}}$$

$$\leq \left( 4d + 4dM_1^2 \mathbb{E}(W_1(\mathbf{x})^2) + 4D_\epsilon(1 + M_2 \mathbb{E}(W_2(\mathbf{x}))) + 4D_\epsilon M_1^2 \mathbb{E}((1 + M_2 W_2(\mathbf{x}))W_1(\mathbf{x})^2) \right)^{\frac{1}{4}}$$

$$\leq \left( 4d + 8dM_1^2(1 + M_2 W_2(\boldsymbol{x})) + 4D_\epsilon(1 + M_2(1 + M_2 W_2(\boldsymbol{x}))) + 4D_\epsilon M_1^2(2 + 2M_2 W_2(\boldsymbol{x}) + 4M_2(1 + M_4 W_4(\boldsymbol{x}))) \right)^{\frac{1}{4}}$$

$$\leq g(\boldsymbol{x}),$$

with $g(\boldsymbol{x})^4 = 4d + 8dM_1^2(1 + M_2 W_2(\boldsymbol{x})) + 4D_\epsilon(1 + M_2(1 + M_2 W_2(\boldsymbol{x}))) + 4D_\epsilon M_1^2(2 + 2M_2 W_2(\boldsymbol{x}) + 4M_2(1 + M_4 W_4(\boldsymbol{x})))$.

By summing the previous inequality, we have

$$\sum_{j=0}^{k-1} D_0 B_a \rho_C^{jm} \delta^{\frac{1}{4}} V(\boldsymbol{x}) \mathbb{E}_{\mathbf{x} \sim \delta_{\boldsymbol{x}}(R^1_{\epsilon,\frac{1}{m}})^{(k-j-1)m}} \left( \sqrt{G(\mathbf{x})} \right) \leq \frac{g(\boldsymbol{x}) V(\boldsymbol{x}) D_0 B_a}{1 - \rho_C^m} \delta^{\frac{1}{4}}$$

$$\leq \frac{g(\boldsymbol{x}) V(\boldsymbol{x}) D_0 B_a}{1 - \rho_C^{\frac{1}{\delta}}} \delta^{\frac{1}{4}}.$$

We denote $D_1 = \frac{g(\boldsymbol{x}) V(\boldsymbol{x}) D_0 B_a}{1 - \rho_C^{\frac{1}{\delta}}}$. We have prove that

$$\sum_{j=0}^{k-1} D_0 B_a \rho_C^{jm} \delta^{\frac{1}{4}} V(\mathbf{x}) \mathbb{E}_{\mathbf{x} \sim \delta_{\boldsymbol{x}}(R^1_{\epsilon,\frac{1}{m}})^{(k-j-1)m}} \left( \sqrt{G(\mathbf{x})} \right) \leq D_1 \delta^{\frac{1}{4}}. \tag{24}$$

Then we study the convergence of the term $\frac{1}{m} \sum_{l=0}^{m-1} \mathbb{E} \left( ||b_\epsilon^1(\boldsymbol{x}_{l,(k-j-1)m}^1) - b_\epsilon^2(\boldsymbol{x}_{l,(k-j-1)m}^1)||^2 \right)$. We denote $\mathbf{y} \sim \pi_{\epsilon,\delta}^1$ a random variable. By a similar computation than in paragraph D.1.3, we have

$$\left| \frac{1}{m} \sum_{l=0}^{m-1} \mathbb{E} \left( ||b_\epsilon^1(\boldsymbol{x}_{l,(k-j-1)m}^1) - b_\epsilon^2(\boldsymbol{x}_{l,(k-j-1)m}^1)||^2 \right) - \mathbb{E} \left( ||b_\epsilon^1(\mathbf{y}) - b_\epsilon^2(\mathbf{y})||^2 \right) \right|$$

$$\leq \frac{1}{m} \sum_{l=0}^{m-1} 8DL_b(1 + M_1 W_1(\boldsymbol{x})) \mathbb{E} \left( ||\boldsymbol{x}_{l,(k-j-1)m}^1 - \mathbf{y}||^2 \right).$$

Moreover, using Equation 23 :

$$\mathbb{E} \left( ||\boldsymbol{x}_{l,(k-j-1)m}^1 - \mathbf{y}||^2 \right) \leq \mathbb{E} \left( ||\boldsymbol{x}_{l,(k-j-1)m}^1 - \mathbf{y}||^2 \mathbf{1}_{\boldsymbol{x}_{l,(k-j-1)m}^1 \neq \mathbf{y}} \right)$$

$$\leq \mathbb{E} \left( ||\boldsymbol{x}_{l,(k-j-1)m}^1 - \mathbf{y}||^2 \right)^{\frac{1}{2}} \mathbb{E} \left( \mathbf{1}_{\boldsymbol{x}_{l,(k-j-1)m}^1 \neq \mathbf{y}} \right)^{\frac{1}{2}}$$

$$\leq \mathbb{E} \left( ||\boldsymbol{x}_{l,(k-j-1)m}^1 - \mathbf{y}||^2 \right)^{\frac{1}{2}} ||\pi_{\epsilon,\delta}^1 - \delta_{\boldsymbol{s}}(R_{\epsilon,\delta}^1)^{(k-j-1)m+l}||_{TV}^{\frac{1}{2}}$$

$$\leq 2M_1 W_1(\boldsymbol{x}) B_a \rho_C^{\frac{(k-j-1)m+l}{2}} V^2(\boldsymbol{x}).$$

By using the two previous computations, we obtain

$$
\left| \frac{1}{m} \sum_{l=0}^{m-1} \mathbb{E}\left( ||b_\epsilon^1(\boldsymbol{x}_{l,(k-j-1)m}^1) - b_\epsilon^2(\boldsymbol{x}_{l,(k-j-1)m}^1)||^2 \right) - \mathbb{E}\left( ||b_\epsilon^1(\mathbf{y}) - b_\epsilon^2(\mathbf{y})||^2 \right) \right|
$$

$$
\leq \frac{1}{m} \sum_{l=0}^{m-1} 16 D L_b (1 + M_1 W_1(\boldsymbol{x})) M_1 W_1(\boldsymbol{x}) B_a \rho_C^{\frac{(k-j-1)m+l}{2}} V^2(\boldsymbol{x})
$$

$$
\leq \frac{16 D L_b M_1 B_a}{1 - \sqrt{\rho_C}} (1 + M_1 W_1(\boldsymbol{x})) W_1(\boldsymbol{x}) V^2(\boldsymbol{x}) \rho_C^{\frac{(k-j-1)m}{2}}.
$$

Then looking at the accumulation of these errors, we have

$$
\left| \sum_{j=0}^{k-1} \sqrt{2} B_a \rho_C^{jm} V(\boldsymbol{x}) \left( \sqrt{\frac{1}{m} \sum_{l=0}^{m-1} \mathbb{E}\left( ||b_\epsilon^1(\boldsymbol{x}_{l,(k-j-1)m}^1) - b_\epsilon^2(\boldsymbol{x}_{l,(k-j-1)m}^1)||^2 \right)} - \sqrt{\mathbb{E}\left( ||b_\epsilon^1(\mathbf{y}) - b_\epsilon^2(\mathbf{y})||^2 \right)} \right) \right|
$$

$$
\leq \frac{4\sqrt{2 D L_b M_1 B_a} B_a}{\sqrt{1 - \sqrt{\rho_C}}} \sqrt{(1 + M_1 W_1(\boldsymbol{x})) W_1(\boldsymbol{x}) V^2(\boldsymbol{x})} \sum_{j=0}^{k-1} \rho_C^{jm} \rho_C^{\frac{(k-j-1)m}{4}}
$$

$$
\leq \frac{4\sqrt{2 D L_b M_1 B_a} B_a}{\sqrt{1 - \sqrt{\rho_C}}} \sqrt{(1 + M_1 W_1(\boldsymbol{x})) W_1(\boldsymbol{x}) V^2(\boldsymbol{x})} k \rho_C^{\frac{(k-1)m}{4}}.
$$

We denote $D_2 = \frac{4\sqrt{2 D L_b M_1 B_a} B_a}{\sqrt{1 - \sqrt{\rho_C}}} \sqrt{(1 + M_1 W_1(\boldsymbol{x})) W_1(\boldsymbol{x}) V^2(\boldsymbol{x})}$, we have demonstrated that the accumulation of error is controlled by

$$
\left| \sum_{j=0}^{k-1} \sqrt{2} B_a \rho_C^{jm} V(\boldsymbol{x}) \left( \sqrt{\frac{1}{m} \sum_{l=0}^{m-1} \mathbb{E}\left( ||b_\epsilon^1(\boldsymbol{x}_{l,(k-j-1)m}^1) - b_\epsilon^2(\boldsymbol{x}_{l,(k-j-1)m}^1)||^2 \right)} - \sqrt{\mathbb{E}\left( ||b_\epsilon^1(\mathbf{y}) - b_\epsilon^2(\mathbf{y})||^2 \right)} \right) \right| \leq D_2 k \rho_C^{\frac{(k-1)m}{4}}
$$

$$(25)$$

Hence, we have

$$
\sum_{j=0}^{k-1} \sqrt{2} B_a \rho_C^{jm} V(\mathbf{x}) \sqrt{\mathbb{E}\left( ||b_\epsilon^1(\mathbf{y}) - b_\epsilon^2(\mathbf{y})||^2 \right)} = \frac{\sqrt{2} B_a V(\mathbf{x})}{1 - \rho_C^m} \sqrt{\mathbb{E}_{\mathbf{y} \sim \pi_{\epsilon,\delta}^1}\left( ||b_\epsilon^1(\mathbf{y}) - b_\epsilon^2(\mathbf{y})||^2 \right)}
$$

$$
\leq \frac{\sqrt{2} B_a V(\mathbf{x})}{1 - \rho_C^{\frac{1}{\delta}}} \sqrt{\mathbb{E}_{\mathbf{y} \sim \pi_{\epsilon,\delta}^1}\left( ||b_\epsilon^1(\mathbf{y}) - b_\epsilon^2(\mathbf{y})||^2 \right)}.
$$

By denoting $D_3 = \frac{\sqrt{2} B_a V(\boldsymbol{x})}{1 - \rho_C^{\frac{1}{\delta}}}$, we get

$$
\sum_{j=0}^{k-1} \sqrt{2} B_a \rho_C^{jm} V(\mathbf{x}) \sqrt{\mathbb{E}\left( ||b_\epsilon^1(\mathbf{y}) - b_\epsilon^2(\mathbf{y})||^2 \right)} \leq D_3 \sqrt{\mathbb{E}_{\mathbf{y} \sim \pi_{\epsilon,\delta}^1}\left( ||b_\epsilon^1(\mathbf{y}) - b_\epsilon^2(\mathbf{y})||^2 \right)}. \tag{26}
$$

Combining Equation 24, Equation 25 and Equation 26, we get

$$
||\delta_{\boldsymbol{x}}(R_{\epsilon,\frac{1}{m}}^1)^{km} - \delta_{\boldsymbol{x}}(R_{\epsilon,\frac{1}{m}}^2)^{km}||_V \leq D_1 \delta^{\frac{1}{4}} + D_2 k \rho_C^{\frac{(k-1)m}{2}} + D_3 \sqrt{\mathbb{E}_{\mathbf{y} \sim \pi_{\epsilon,\delta}^1}\left( ||b_\epsilon^1(\mathbf{y}) - b_\epsilon^2(\mathbf{y})||^2 \right)}. \tag{27}
$$

### D.1.4 Demonstration of $TV$-distance bound of Theorem 1

We go back in the initial decomposition (Equation 11). By taking $a = \max(a_1, a_2) \in \mathbb{R}^+$ and $\rho = \max(\rho_1, \rho_2) \in ]0, 1[$ in Equation D.1.2, and using Equation 27, for $k \in \mathbb{N}$ and $\delta = \frac{1}{m}$, we get

$$
||\pi_{\epsilon,\delta}^1 - \pi_{\epsilon,\delta}^2||_{TV} \leq ||\delta_{\boldsymbol{x}}(R_{\epsilon,\delta}^1)^{km} - \pi_{\epsilon,\delta}^1||_{TV} + ||\delta_{\boldsymbol{x}}(R_{\epsilon,\delta}^1)^{km} - \delta_{\boldsymbol{x}}(R_{\epsilon,\delta}^2)^{km}||_{TV} + ||\delta_{\boldsymbol{x}}(R_{\epsilon,\delta}^2)^{km} - \pi_{\epsilon,\delta}^2||_{TV}
$$

$$
\leq a\rho^{km\delta} \left( 2V^2(\boldsymbol{x}) + \int_{\mathbb{R}^d} V^2(\tilde{\boldsymbol{x}})(\pi_{\epsilon,\delta}^1 + \pi_{\epsilon,\delta}^2)(d\tilde{\boldsymbol{x}}) \right) + D_1 \delta^{\frac{1}{4}} + D_2 k \rho_C^{\frac{(k-1)m}{2}} + D_3 \sqrt{\mathbb{E}_{\mathbf{y} \sim \pi_{\epsilon,\delta}^1}\left( ||b_\epsilon^1(\mathbf{y}) - b_\epsilon^2(\mathbf{y})||^2 \right)}.
$$

By taking, $k \mapsto +\infty$, we have for $\delta = \frac{1}{m} < \bar{\delta}$ :

$$||\pi^1_{\epsilon,\delta} - \pi^2_{\epsilon,\delta}||_{TV} \leq D_3 \sqrt{\mathbb{E}_{\mathbf{y} \sim \pi^1_{\epsilon,\delta}} (||b^1_\epsilon(\mathbf{y}) - b^2_\epsilon(\mathbf{y})||^2)} + D_1 \delta^{\frac{1}{4}}.$$

The parameter $\boldsymbol{x}$ (appear inside the constants) is free in the last inequality. We choose $\boldsymbol{x} = 0$. It gives that there exist two constants $A_0, B_0 \geq 0$ defined by :

- $A_0 = D_3$
- $B_0 = D_1$

such as for $\delta = \frac{1}{m} < \bar{\delta} = \frac{1}{3} \left( L + \frac{M+1}{\epsilon} + \frac{1}{\lambda} \right)^{-1}$ :

$$||\pi^1_{\epsilon,\delta} - \pi^2_{\epsilon,\delta}||_{TV} \leq A_0 d_1(b^1_\epsilon, b^2_\epsilon) + B_0 \delta^{\frac{1}{4}}.$$

With the posterior-$L_2$ defined in 4.1. This demonstrated the Theorem 1.

### D.1.5 DEMONSTRATION OF $\mathbf{W}_1$-DISTANCE BOUND OF THEOREM 1

Let $\mathbf{x}$ and $\mathbf{y}$ be two random variable of laws $\mathbf{x} \sim \pi^1_{\epsilon,\delta}$ and $\mathbf{y} \sim \pi^2_{\epsilon,\delta}$.

By the definition of the Wasserstein distance (Equation 8) and the Cauchy-Schwarz inequality

$$
\begin{aligned}
\mathbf{W}_1(\pi^1_{\epsilon,\delta}, \pi^2_{\epsilon,\delta}) &\leq \mathbb{E}[||\mathbf{x} - \mathbf{y}||] \\
&\leq \mathbb{E}[||\mathbf{x} - \mathbf{y}||\mathbf{1}_{\mathbf{x} \neq \mathbf{y}}] \\
&\leq \mathbb{E}[(||\mathbf{x}|| + ||\mathbf{y}||)\mathbf{1}_{\mathbf{x} \neq \mathbf{y}}] \\
&\leq \mathbb{E}[(||\mathbf{x}|| + ||\mathbf{y}||)^2]^{1/2} \mathbb{E}[\mathbf{1}_{\mathbf{x} \neq \mathbf{y}}]^{1/2} \\
&\leq \sqrt{2} \mathbb{E}[||\mathbf{x}||^2 + ||\mathbf{y}||^2]^{1/2} \mathbb{E}[\mathbf{1}_{\mathbf{x} \neq \mathbf{y}}]^{1/2} \\
&\leq \sqrt{2} \mathbb{E}[||\mathbf{x}||^2 + ||\mathbf{y}||^2]^{1/2} ||\pi^1_{\epsilon,\delta} - \pi^2_{\epsilon,\delta}||^{1/2}_{TV}.
\end{aligned}
$$

Using Cauchy-Schwarz inequality and the moment bound (Equation 14)

$$
\begin{aligned}
\mathbf{W}_1(\pi^1_{\epsilon,\delta}, \pi^2_{\epsilon,\delta}) &\leq \mathbb{E}[||\mathbf{x} - \mathbf{y}||] \\
&\leq 2\sqrt{M_2} ||\pi^1_{\epsilon,\delta} - \pi^2_{\epsilon,\delta}||^{1/2}_{TV} \\
&\leq 2\sqrt{M_2} \left( A_0 d_1(b^1_\epsilon, b^2_\epsilon) + B_0 \delta^{\frac{1}{4}} \right)^{\frac{1}{2}} \\
&\leq 2\sqrt{M_2}\sqrt{A_0} \left( d_1(b^1_\epsilon, b^2_\epsilon) \right)^{\frac{1}{2}} + 2\sqrt{M_2}\sqrt{B_0} \delta^{\frac{1}{8}}.
\end{aligned}
$$

By the first part of Theorem 1, the total variation distance between the two invariant distributions can be bound. We denote the constant $A_1 = 2\sqrt{M_2}\sqrt{A_0}$ and $B_1 = 2\sqrt{M_2}\sqrt{B_0}$. The following theorem have been prove, which shows the second part of the Theorem 1.

### D.1.6 ANALYSIS OF THE CONSTANT $A_0$, $B_0$, $A_1$ AND $B_1$

In this section, we analyse the dependency of the constant $A_0$ and $B_0$. first of all by construction, these constants are independent of $\delta \in ]0, \bar{\delta}]$, with $\bar{\delta} = \frac{1}{3} \left( L + \frac{M+1}{\epsilon} + \frac{1}{\lambda} \right)^{-1}$. Also these constant are dependent of the inverse problem observation $\boldsymbol{y}$, in fact they depend of the targeted posterior distribution (through the constant $B_a$, $\rho_C$).

**Study of $A_0$**
The definition of $A_0$ is

$$A_0 = D_3 = \frac{\sqrt{2} B_a}{1 - \rho_C^{\frac{1}{\delta}}}$$

Where $B_a$ and $\rho_C$ are the constant defined in the Equation 23. Bortoli & Durmus (2020) has proved that $\rho_C$ is independent of the dimension $d$ and moreover this rate is closed to the optimal rate thank

to Eberle (2016, Lemma 2). However Bortoli & Durmus (2020) has also prove that the constant $B_a$ has a polynomial dependent on the dimension. So there exists $r \geq 0$ and $C_r > 0$ such that

$$B_a \underset{d \to +\infty}{\sim} C_r d^r. \tag{28}$$

Thus $A_0$ has a polynomial dependency on the dimension

$$A_0 \underset{d \to +\infty}{\sim} \frac{\sqrt{2} C_r}{1 - \rho_C^{\frac{1}{\delta}}} d^r. \tag{29}$$

**Study of $B_0$**

The definition of $B_0$ is

$$B_0 = D_1 = \frac{g(0) D_0 B_a}{1 - \rho_C^{\frac{1}{\delta}}}$$
$$= (1 - \rho_C^{\frac{1}{\delta}})^{-1} \left( \left( 4d + 8d M_1^2 (1 + M_2) + 4 D_\epsilon (1 + M_2(1 + M_2)) + 4 D_\epsilon M_1^2 (2 + 2M_2 + 4M_2(1 + M_4)) \right)^{\frac{1}{4}} \sqrt{16 D_\epsilon L_b} B_a \right).$$

Thanks to the dependency of $B_a$ to the dimension in Equation 28, $B_0$ has the dependency

$$B_0 \underset{d \to +\infty}{\sim} (1 - \rho_C^{\frac{1}{\delta}})^{-1} \left( 4 + 8 M_1^2 (1 + M_2) \right)^{\frac{1}{4}} \sqrt{16 D_\epsilon L_b} C_r d^{r + \frac{1}{4}}. \tag{30}$$

Recall that $M_1$ and $M_2$ are the moment upper-bound, $L_b$ a Lipschitz constant of $b_\epsilon^i$ and $D_\epsilon = L_b + \|b_\epsilon(0)\|$. These quantity has no dependency on the dimension.

**Study of $A_1$**

The definition of $A_1$ is

$$A_1 = 2\sqrt{M_2}\sqrt{A_0}.$$

Using Equation 29, $A_1$ has the dimension dependence

$$A_1 \underset{d \to +\infty}{\sim} 2\sqrt{M_2} \sqrt{\frac{\sqrt{2} C_r}{1 - \rho_C^{\frac{1}{\delta}}}} d^{\frac{r}{2}}.$$

**Study of $B_1$**

The definition of $B_1$ is

$$B_1 = 2\sqrt{M_2}\sqrt{B_0}.$$

Using Equation 30, $B_1$ has the following dimension dependence

$$B_1 \underset{d \to +\infty}{\sim} 2\sqrt{M_2} \sqrt{(1 - \rho_C^{\frac{1}{\delta}})^{-1} \left( 4 + 8 M_1^2 (1 + M_2) \right)^{\frac{1}{4}} \sqrt{16 D_\epsilon L_b} C_r} d^{\frac{r}{2} + \frac{1}{8}}.$$

### D.1.7 ADDITIONAL RESULTS

With a similar demonstration than before, we can obtain bound on the MMSE estimator or the standard deviation of the Markov chain instead of the distribution. In fact, in practice, we mainly look at the MMSE (mean of the Markov chain) and the standard deviation of the Markov chain (as a confidence map).

**MMSE error bound**     With the same **x** and **y** random variable than above :

$$\|\mathbb{E}[\mathbf{x}] - \mathbb{E}[\mathbf{y}]\| \leq \mathbb{E}[\|\mathbf{x} - \mathbf{y}\|]$$
$$\leq 2\sqrt{M_2}\|\pi_{\epsilon,\delta}^1 - \pi_{\epsilon,\delta}^2\|_{TV}^{1/2}.$$

The expectation of $\pi_{\epsilon,\delta}^1$ is the MMSE estimator of the inverse problem with the prior defined by the first denoiser. If we name, for $i \in \{1, 2\}$, $\hat{\boldsymbol{x}}_{MMSE,i} = \mathbb{E}_{\mathbf{x} \sim \pi_{\epsilon,\delta}^i}[\mathbf{x}]$. The following proposition has been prove (with $A_5 = A_1, B_5 = B_1$).

**Proposition 1.1.** It $A_5, B_5 \in \mathbb{R}^+$, such as for $\delta \in ]0, \bar{\delta}]$ :

$$\|\hat{\boldsymbol{x}}_{MMSE,1} - \hat{\boldsymbol{x}}_{MMSE,2}\| \leq A_5 \left(d_1(b_\epsilon^1, b_\epsilon^2)\right)^{\frac{1}{2}} + B_5 \delta^{\frac{1}{8}}. \tag{31}$$

**Standard deviation error bound**     To control the difference between variance of the samples

$$\begin{aligned}
|\mathbf{Var}[\mathbf{x}] - \mathbf{Var}[\mathbf{y}]| &= \left|\mathbb{E}[\|\mathbf{x}\|^2] - \|\mathbb{E}[\mathbf{x}]\|^2 - \mathbb{E}[\|\mathbf{y}\|^2] + \|\mathbb{E}[\mathbf{y}]\|^2\right| \\
&\leq \left|\mathbb{E}[\|\mathbf{x}\|^2] - \mathbb{E}[\|\mathbf{y}\|^2]+\right| + \left|\|\mathbb{E}[\mathbf{y}]\|^2 - \|\mathbb{E}[\mathbf{x}]\|^2\right| \\
&\leq \mathbb{E}\big[\big|\|\mathbf{x}\|^2 - \|\mathbf{y}\|^2\big|\, \mathbf{1}_{\mathbf{x} \neq \mathbf{y}}\big] + \mathbb{E}[\|\mathbf{x} - \mathbf{y}\|]\,(\|\mathbb{E}[\mathbf{x}]\| + \|\mathbb{E}[\mathbf{y}]\|) \\
&\leq (\sqrt{2M_4} + 2\sqrt{2M_2}M_1)\|\pi_{\epsilon,\delta}^1 - \pi_{\epsilon,\delta}^2\|_{TV}^{1/2}.
\end{aligned}$$

We denote $A_6 = (\sqrt{2M_4} + 2\sqrt{2M_2}M_1)\sqrt{A_0}$ and $B_6 = B_0$, the following proposition has been proved.

**Proposition 1.2.** There exist $A_6, B_6 \in \mathbb{R}^+$, , such that for $\delta \in ]0, \bar{\delta}]$ :

$$\left|\mathbf{Var}[\pi_{\epsilon,\delta}^1] - \mathbf{Var}[\pi_{\epsilon,\delta}^2]\right| \leq A_6 \left(d_1(b_\epsilon^1, b_\epsilon^2)\right)^{\frac{1}{2}} + B_6 \delta^{\frac{1}{8}}. \tag{32}$$

# E   COROLLARY DEMONSTRATIONS

## E.1   DEMONSTRATION OF COROLLARY 1.1

If the forward problem is the same for the two Markov chain, the only shift is on the denoiser. Then, the distance between the two drift is

$$d_1(b_\epsilon^1, b_\epsilon^2) = \frac{1}{\epsilon}\sqrt{\mathbb{E}_{X \sim \pi_{\epsilon,\delta}^1}\left(\|D_\epsilon^1(X) - D_\epsilon^2(X)\|^2\right)} = \frac{1}{\epsilon}d_1(D_\epsilon^1, D_\epsilon^2).$$

Hence we have

$$\|\pi_{\epsilon,\delta}^1 - \pi_{\epsilon,\delta}^2\|_{TV} \leq \frac{A_0}{\epsilon}d_1(D_\epsilon^1, D_\epsilon^2) + B_0 \delta^{\frac{1}{4}}.$$

So the result 1.1 has been demonstrated with $A_2 = \frac{A_0}{\epsilon}$ and $B_2 = B_0$.

## E.2   DEMONSTRATION OF COROLLARY 1.2

If $D_\epsilon^1 = D_\epsilon^\star$, the exact MMSE denoiser. Some more assumption are needed on the prior. We defi,e

$$g_\epsilon(x_1|x_2) = p^\star(x_1) \exp\left(-\frac{\|x_2 - x_1\|^2}{2\epsilon}\right) \Big/ \int_{\mathbb{R}^d} p^\star(\tilde{\boldsymbol{x}}) \exp\left(-\frac{\|x_2 - \tilde{\boldsymbol{x}}\|^2}{2\epsilon}\right) d\tilde{\boldsymbol{x}}.$$

**Assumption 5.** For any $\epsilon > 0$, there exists $K_\epsilon \geq 0$ such that $\forall \boldsymbol{x} \in \mathbb{R}^d$,

$$\int_{\mathbb{R}^d} \|\tilde{\boldsymbol{x}} - \int_{\mathbb{R}^d} \tilde{\boldsymbol{x}}' g_\epsilon(\tilde{\boldsymbol{x}}'|\boldsymbol{x}) d\tilde{\boldsymbol{x}}'\|^2 g_\epsilon(\tilde{\boldsymbol{x}}|\boldsymbol{x}) d\tilde{\boldsymbol{x}} \leq K_\epsilon.$$

And

$$\int_{\mathbb{R}^d} (1 + \|\tilde{\boldsymbol{x}}\|^4) p_\epsilon^\star(\tilde{\boldsymbol{x}}) d\tilde{\boldsymbol{x}} < +\infty.$$

Using Assumptions 1-5 and (Laumont et al., 2022, Proposition 6), applied with the exact MMSE denoiser ($M_R = 0$ and $R = +\infty$). If $V(\boldsymbol{x}) = 1 + \|x\|^2$, for $\epsilon \leq \epsilon_0$, and $\lambda > 0$ such that $2\lambda(L_y + \frac{L}{\epsilon} - \min(m, 0)) \leq 1$ and $\bar{\delta} = \frac{1}{3}(L_y + \frac{L}{\epsilon} + \frac{1}{\lambda})^{-1}$. There exists $D_4 \geq 0$ such that for $R_C > 0$ such that $\bar{B}(0, R_C) \subset \mathbb{S}$, there exists $D_{5,C} \geq 0$ such that $\forall \delta \in\, ]0, \bar{\delta}]$ :

$$\|\pi^1_{\epsilon,\delta} - p_\epsilon(\cdot|y)\|_V \leq D_4 R_C^{-1} + D_{5,C}\sqrt{\delta}.$$

Due to $\|\cdot\|_{TV} \leq \|\cdot\|_V$, we have

$$\|\pi^1_{\epsilon,\delta} - p_\epsilon(\cdot|y)\|_{TV} \leq D_4 R_C^{-1} + D_{5,C}\sqrt{\delta}.$$

Using the triangular inequality and Corollary 1.1, for $0 < \delta < \min\left(\bar{\delta}, 1\right)$

$$\begin{aligned}
\|p_\epsilon(\cdot|y) - \pi^2_{\epsilon,\delta}\|_{TV} &\leq \|\pi^1_{\epsilon,\delta} - \pi^2_{\epsilon,\delta}\|_{TV} + \|\pi^1_{\epsilon,\delta} - p_\epsilon(\cdot|y)\|_{TV} \\
&\leq A_2 d_1(D^\star_\epsilon, D^2_\epsilon) + B_2 \delta^{\frac{1}{4}} + D_4 R_C^{-1} + D_{5,C}\sqrt{\delta} \\
&\leq A_2 d_1(D^\star_\epsilon, D^2_\epsilon) + (B_2 + D_{5,C})\delta^{\frac{1}{4}} + D_4 R_C^{-1}.
\end{aligned}$$

Finally, the corollary 1.2 has been demonstrated with $A_3 = A_1$, $B_3 = (B_2 + D_{5,C})$ and $C_3 = D_4$.

### E.3  DEMONSTRATION OF COROLLARY 1.3

In this case, the denoiser is fix but there is a shift on the forward problem.

$$D^1_\epsilon = D^2_\epsilon$$
$$\nabla \log p^1(\boldsymbol{y}|\cdot) \neq \nabla \log p^2(\boldsymbol{y}|\cdot).$$

Because of the forward model 2, the likelihood has the form

$$\nabla_{\boldsymbol{x}} \log p^i(\boldsymbol{y}|\boldsymbol{x}) = -\nabla_{\boldsymbol{x}} \frac{\|\boldsymbol{y} - \boldsymbol{A}_i(\boldsymbol{x})\|^2}{2\sigma^2} = \frac{1}{\sigma^2} \boldsymbol{A}_i^T(\boldsymbol{y} - \boldsymbol{A}_i\boldsymbol{x}).$$

Hence, we have

$$\begin{aligned}
b^1_\epsilon(\boldsymbol{x}) - b^2_\epsilon(\boldsymbol{x}) &= \nabla_{\boldsymbol{x}} \log p^1(\boldsymbol{y}|\boldsymbol{x}) - \nabla_{\boldsymbol{x}} \log p^2(\boldsymbol{y}|\boldsymbol{x}) = \frac{1}{\sigma^2}(\boldsymbol{A}_1^T(\boldsymbol{y} - \boldsymbol{A}_1\boldsymbol{x}) - \boldsymbol{A}_2^T(\boldsymbol{y} - \boldsymbol{A}_2\boldsymbol{x})) \\
&= \frac{1}{\sigma^2}((\boldsymbol{A}_1 - \boldsymbol{A}_2)^T\boldsymbol{y} - (\boldsymbol{A}_1^T\boldsymbol{A}_1 - \boldsymbol{A}_2^T\boldsymbol{A}_2)\boldsymbol{x}).
\end{aligned}$$

Moreover, we can develop

$$\boldsymbol{A}_1^T\boldsymbol{A}_1 - \boldsymbol{A}_2^T\boldsymbol{A}_2 = \boldsymbol{A}_1^T\boldsymbol{A}_1 - \boldsymbol{A}_2^T\boldsymbol{A}_1 + \boldsymbol{A}_2^T\boldsymbol{A}_1 - \boldsymbol{A}_2^T\boldsymbol{A}_2 = (\boldsymbol{A}_1 - \boldsymbol{A}_2)^T\boldsymbol{A}_1 + \boldsymbol{A}_2^T(\boldsymbol{A}_1 - \boldsymbol{A}_2).$$

So, with the operator norm, note $\|\cdot\|$, on matrix associated with the Euclidean norm, the following inequality holds

$$\begin{aligned}
\|b^1_\epsilon(\boldsymbol{x}) - b^2_\epsilon(\boldsymbol{x})\| &= \frac{1}{\sigma^2}\left\|(\boldsymbol{A}_1 - \boldsymbol{A}_2)^T\boldsymbol{y} - (\boldsymbol{A}_1 - \boldsymbol{A}_2)^T\boldsymbol{A}_1\boldsymbol{x} - \boldsymbol{A}_2^T(\boldsymbol{A}_1 - \boldsymbol{A}_2)\boldsymbol{x}\right\| \\
&\leq \|\boldsymbol{A}_1 - \boldsymbol{A}_2\|\left(\|\boldsymbol{y}\| + (\|\boldsymbol{A}_1\| + \|\boldsymbol{A}_2\|)\|\boldsymbol{x}\|\right).
\end{aligned}$$

Hence, we have

$$\begin{aligned}
\|b^1_\epsilon(\boldsymbol{x}) - b^2_\epsilon(\boldsymbol{x})\|^2 &\leq \|\boldsymbol{A}_1 - \boldsymbol{A}_2\|^2\left(\|\boldsymbol{y}\| + (\|\boldsymbol{A}_1\| + \|\boldsymbol{A}_2\|)\|\boldsymbol{x}\|\right)^2 \\
&\leq 2\|\boldsymbol{A}_1 - \boldsymbol{A}_2\|^2\left(\|\boldsymbol{y}\|^2 + (\|\boldsymbol{A}_1\| + \|\boldsymbol{A}_2\|)^2\|\boldsymbol{x}\|^2\right).
\end{aligned}$$

If $\boldsymbol{A}_1$ and $\boldsymbol{A}_1$ are in a bound domain. There exists $M$ such that $M \geq \max(\|\boldsymbol{A}_1\|, \|\boldsymbol{A}_2\|)$, it gives :

$$\|b^1_\epsilon(\boldsymbol{x}) - b^2_\epsilon(\boldsymbol{x})\|^2 \leq 2\|\boldsymbol{A}_1 - \boldsymbol{A}_2\|^2\left(\|\boldsymbol{y}\|^2 + 4M^2\|\boldsymbol{x}\|^2\right).$$

Hence, we have

$$\begin{aligned}
d_1(b^1_\epsilon, b^2_\epsilon) &= \sqrt{\mathbb{E}_{\mathbf{x}\sim\pi^1_{\epsilon,\delta}}\left(\|b^1_\epsilon(\mathbf{x}) - b^2_\epsilon(\mathbf{x})\|^2\right)} \\
&\leq \sqrt{2\|\boldsymbol{A}_1 - \boldsymbol{A}_2\|^2\left(\|\boldsymbol{y}\|^2 + 4M^2\mathbb{E}_{\mathbf{x}\sim\pi^1_{\epsilon,\delta}}(\|\mathbf{x}\|^2)\right)} \\
&\leq \sqrt{2}\|\boldsymbol{A}_1 - \boldsymbol{A}_2\|\left(\|\boldsymbol{y}\| + 2M\sqrt{\mathbb{E}_{\mathbf{x}\sim\pi^1_{\epsilon,\delta}}(\|\mathbf{x}\|^2)}\right).
\end{aligned}$$

Using the moment bound (Equation 14), $\sqrt{\mathbb{E}_{\mathbf{x} \sim \pi^1_{\epsilon,\delta}}(\|\mathbf{x}\|^2)} \leq \sqrt{M_2}$.

Then Theorem 1 gives, that $\forall \delta \in ]0, \bar{\delta}]$ :

$$\|\pi^1_{\epsilon,\delta} - \pi^2_{\epsilon,\delta}\|_{TV} \leq A_0 d_1(b^1_\epsilon, b^2_\epsilon) + B_0 \delta^{\frac{1}{4}}.$$

Therefore with $A_4 = \sqrt{2} A_0 \left(\|\boldsymbol{y}\| + 2M\sqrt{M_2}\right) \geq 0$ and $B_4 = B_0$, corollary 1.3 has been proved.

