# OpenReview forum: "Plug-and-Play Posterior Sampling under Mismatched Measurement and Prior Models"
_ICLR.cc/2024/Conference — ICLR 2024 poster_

### Official Review · Reviewer_WmLx · 2023-10-15

**Soundness:** 3 good
**Presentation:** 3 good
**Contribution:** 3 good
**Rating:** 6
**Confidence:** 3

**Summary:**

The paper proposes a theory of the plug-and-play unadjusted Langevin algorithm (PnP-ULA) to solve inverse problems, that builds and improves upon a prior work [1]. Specifically, the main theorem quantifies the distributional error of PnP-ULA under prior shift and the likelihood shift, which are both practical questions in the field of inverse imaging. Numerical experiments including the analytical GMM experiment and the image deblurring experiment solidify the correctness of the theorem.

**Strengths:**

1. The paper is well-written, concise, and clear.

2. The theory given in the paper is solid, with numerical experiments building support for the proposed theorem. The theory is practical and non-vacuous, as mismatch in the prior or in the likelihood happens (at least to a minimal amount) on virtually every application that you can think of.

3. The subject of the paper is well-suited for the conference, given the popularity of diffusion models on solving inverse problems.

**Weaknesses:**

1. The prior mismatch model, and the likelihood mismatch model, are not too realistic.

1-1. For the prior mismatch, it would be more interesting if one could see the effect when the underlying training distributions are different. For example, in the context of medical imaging, [5,6] demonstrated that diffusion models are particularly robust under prior distribution shifts. A discussion would be useful.

Note that I understand why the authors opted for the number of parameters for a DNN when they assumed a mismatch from the perfect MMSE estimator. However, given the current landscape of ML/generative AI, this situation would be easily solvable by more compute, whereas solving the data distribution shift is a much harder and realistic problem.

2. The authors cite [1] as an example of a mismatched imaging forward model. Correct me if I am wrong, but as far as I understand, when using unconditional denoisers as in PnP-ULA, [1] uses the exact forward operator that were used to generate the measurement. I believe references such as [2-4] would be more relevant.



**References**

[1] Güngör, Alper, et al. "Adaptive diffusion priors for accelerated MRI reconstruction." Medical Image Analysis (2023): 102872.

[2] Wirgin, Armand. "The inverse crime." 2004.

[3] Guerquin-Kern, Matthieu, et al. "Realistic analytical phantoms for parallel magnetic resonance imaging." IEEE Transactions on Medical Imaging 31.3 (2011): 626-636.

[4] Blanke, Stephanie E., Bernadette N. Hahn, and Anne Wald. "Inverse problems with inexact forward operator: iterative regularization and application in dynamic imaging." Inverse Problems 36.12 (2020): 124001.

[5] Jalal, Ajil, et al. "Robust compressed sensing mri with deep generative priors." NeurIPS 2021.

[6] Chung, Hyungjin, and Jong Chul Ye. "Score-based diffusion models for accelerated MRI." Medical image analysis 80 (2022): 102479.

**Questions:**

1. In the forward model shift experiment on color image deblurring, what happens if one takes $\sigma > 3$, and taking to the extreme, when one uses a uniform blur kernel?

2. For image deblurring, what happens when you have an anisotropic blur kernel, but you use an isotropic kernel for inference?

3. Two different versions of references are given for [1]

4. It is probably better to cite [2] rather than [3] for score-matching (pg. 2)





**References**

[1] Chung, Hyungjin, et al. "Diffusion posterior sampling for general noisy inverse problems." ICLR 2023.

[2] Vincent, Pascal. "A connection between score matching and denoising autoencoders." Neural computation 23.7 (2011): 1661-1674.

[3] Dhariwal, Prafulla, and Alexander Nichol. "Diffusion models beat gans on image synthesis." NeurIPS 2022

---

> ### Author Response · Authors · 2023-11-18
> **Response to Reviewer WmLx (part 1/2)**
>
> Thank you for your positive and accurate feedback, as well as the careful reading of our paper. In what follows, we answer to the comments and questions of the reviewer in details.
>
>  - The prior mismatch model, and the likelihood mismatch model, are not too realistic.
>     1-1. For the prior mismatch, it would be more interesting if one could see the effect when the underlying training distributions are different. For example, in the context of medical imaging, [5,6] demonstrated that diffusion models are particularly robust under prior distribution shifts. A discussion would be useful.
>     Note that I understand why the authors opted for the number of parameters for a DNN when they assumed a mismatch from the perfect MMSE estimator. However, given the current landscape of ML/generative AI, this situation would be easily solvable by more compute, whereas solving the data distribution shift is a much harder and realistic problem.
>
> Our theoretical contribution can cover both the parameter mismatch and the training data mismatch. Our experiments are designed to assess our bound and provide direct support for Theorem 1 in the context of images, rather than addressing the specific issue of mismatched training distributions.
>
> For the mismatch forward model, our theory cover each type of mismatch. We choose to use Gaussian blur in order to compute in closed-form the quantity $\|A_1 - A_2\|$ in Corollary 1.3. With more complex blur (anisotropic kernel of blur in particular) this quantity is not closed-form anymore.
> For the prior mismatch, we need to construct a set of different denoisers. By playing with the number of parameters, it is clear that the most powerful denoiser can be taken as a reference. In that setting, it was possible for us to construct a range of denoisers with different distances to the reference denoiser.
>
> We also studied the stability to the underlying training distribution. Two datasets $S_0$ and $S_1$ (MRI images and natural images) were mixed to form a set of training datasets $S_t$ (with a proportion $1-t$ of images from $S_0$ and $t$ from $S_1$) and we trained a set of denoisers $D_t$ on these datasets. Considering the MRI image debluring task,  we observed that even with a little proportion (such as $t = 0.1$) of images from $S_1$, the denoiser $D_t$ has an important shift with $D_0$ in the posterior-$L_2$ metric. It was challenging to develop a set of intermediary denoisers for a more precise evaluation of our bound. Therefore we have decided to switch to the current setting, with different numbers of parameters, for which the construction of intermediary denoisers was easier.
>
> After reading [5,6], there are relevant citation in our work. We propose to cite [5,6] in our introduction in the revised manuscript introduction.
>
>  - The authors cite [1] as an example of a mismatched imaging forward model. Correct me if I am wrong, but as far as I understand, when using unconditional denoisers as in PnP-ULA, [1] uses the exact forward operator that were used to generate the measurement. I believe references such as [2-4] would be more relevant.
>
> You are right, thank you for noticing this error. [1] studies a domain shift and not a measurement model shift. We read the $3$ papers you mentioned. These papers are indeed relevant and they are now cited in the revised version.
>
>  - In the forward model shift experiment on color image deblurring, what happens if one takes $\sigma > 3$, and taking to the extreme, when one uses a uniform blur kernel?
>
> This question is now answered in the supplementary materials (.zip) named "supplementary\_experiments.pdf". In this document, one can see that the behavior is similar for $\sigma_1 > 3$ and  $\sigma_1 < 3$. The further away we are from the exact measurement model, the greater the Wasserstein distance. Moreover the dependency seems to be sub-linear, which is consistent with our theoretical result in Corollary 1.3.
>
> In the context of reconstructing an image with an overestimated blur parameter $\sigma_1$, we observe artifacts which look like aliasing. However one can note that the reconstruction is stable for $\sigma_1 \approx 3$.

---

> ### Author Response · Authors · 2023-11-18
> **Response to Reviewer WmLx (part 2/2)**
>
> - For image deblurring, what happens when you have an anisotropic blur kernel, but you use an isotropic kernel for inference?
>
> This question is now answered in supplementary materials (.zip) in "supplementary\_experiments.pdf". We can see that the behavior is more complex with respect to the previous setting. The degradation kernel is Gaussian with a standard deviation of $5$ horizontally and a standard deviation of $3$ vertically. We take an isotropic blur kernel (with a standard deviation $\sigma_1$) to reconstruct the image. The reconstruction obtained with this isotropic kernel is visually worse than the one obtained with the exact forward model. Moreover, when the kernel is under-estimated ($\sigma_1 < 3$), the reconstructed image stays blurred. If $3 < \sigma_1 < 5$, the "ringing" effect starts to occur vertically but not horizontally. Finally for $\sigma_1 > 5$, the "ringing" effect is visible in both directions.
>
>  - Two different versions of references are given for [1]
>  - It is probably better to cite [2] rather than [3] for score-matching (pg. 2)
>
> Thank you for the careful reading of the references. We have corrected the citation and cited Vincent et al in page 2.

---

> > ### Comment · Reviewer_WmLx · 2023-11-22
> >
> > I would like to thank the authors for addressing my concerns. I will keep my score.

---

### Official Review · Reviewer_rc8y · 2023-10-30

**Soundness:** 3 good
**Presentation:** 3 good
**Contribution:** 3 good
**Rating:** 6
**Confidence:** 2

**Summary:**

The paper studies the error bound of the plug-and-play unadjusted Langevin algorithm (PnP-ULA) under mismatched posterior distribution owing to mismatched data fidelity and/or denoiser. After rigorously deriving their main theoretical results, they provide some numerical experiments with simple settings to further support their theory.

**Strengths:**

- Sections 1–4 are generally clearly written. The readers can get what the authors try to convey without diving into the mathematical details.
- The quantification of the error bound for PnP-ULA under a mismatched posterior distribution is of theoretical importance.

**Weaknesses:**

- The section associated with the numerical experiment is hard to dig out. Particularly, it's not easy to understand how and why the proposed setting can be adopted to validate the theoretical corollary.
- As claimed by the authors, "our results can be seen as a PnP counterpart of existing results on diffusion models.", which therefore weakens the novelty of this paper.
- It seems like the theoretical results drawn rely on "oracle" information that is unavailable in practice. So the practical use of this theoretical tool is largely unclear for me.

**Questions:**

See above

____
After rebuttal: the authors addressed my concerns, thus I raise my score.

---

> ### Author Response · Authors · 2023-11-18
> **Response to Reviewer rc8y (part 1/2)**
>
> Thank you for your positive feedback and careful reading of our paper. Below we provide detailed responses to your comments.
>
>  - The section associated with the numerical experiment is hard to dig out. Particularly, it's not easy to understand how and why the proposed setting can be adopted to validate the theoretical corollary.
>
> Our experiments are developed to support our main theoretical results derived from Theorem 1. We provide one experiment for each corollary.
>
> Both Figure 1 left (2D GMM) and right (grayscale image denoising) illustrate Theorem 1 with respect to the Wasserstein distance and the posterior-$L_2$ pseudometric. This is due to the fact that the $TV$-distance cannot be easily computed because of the discretization of samples.  Additionally, the constants $A_0$, $B_0$, $A_1$ and $B_1$ are not tractable. The high correlation between the quantity $\mathbf{W_1}(\pi_{\epsilon, \delta}^1, \pi_{\epsilon, \delta}^2)$ and $d_1(b_{\epsilon}^1,b_{\epsilon}^2)$ in Figure 1 indicates that the drift shift can accurately reflected the samples shift, which is our theoretical result.
>
> More specifically, for the 2D GMM experiment depicted in Figure 1 (left), we test the bound of Corollary 1.2. The denoisers used in this experiment are illustrated in Figure 2. With a GMM prior, the posterior is in closed-form, making the Wasserstein distance tractable. Similarly, the posterior-$L2$ is also tractable. In other words, both sides of the inequality in Corollary 1.2 are tractable, excepting constants $A_3$ and $B_3$.
>
> The experiment on gray-scale images in Figure 1 (right) corresponds to Corollary 1.1. Since the exact MMSE denoiser and the posterior are not tractable for images, we learn a powerful denoiser $D_{\epsilon}^1$ (with a large number of parameter) and a set of denoisers $D_{\epsilon}^2$ with fewer parameters. Using two samples-one computed with $D_{\epsilon}^1$ and the other with $D_{\epsilon}^2$-we can compute both sides of the bound of Corollary 1.1, except for constants $A_2$, $B_2$. Our numerical result reveals a high correlation, indicating that our theorem is validated for images.
>
> The experiment on color images provides in Figure 4 is made to verify Corollary 1.3. We compute two Markov Chains, one with the exact forward model (involving a Gaussian blur $\sigma^*$) and the other with a mismatched forward model (with a Gaussian blur $\sigma_1$). The Wasserstein distance between the two sampling is then compared to the difference between the two standard deviations, which is exactly $\|A_1 - A_2\|$ for a Gaussian blur. In this case, the quantities are still positively correlated in terms of qualitative behavior, as defined by our bounds.
>
> Therefore, our three experiments give practical illustrations to the three corollaries demonstrated in the theoretical part.
>
>  - As claimed by the authors, ``our results can be seen as a PnP counterpart of existing results on diffusion models.", which therefore weakens the novelty of this paper.
>
> We have remove this statement in the revised manuscript.
> Our work provides the first theoretical stability analysis to forward-model or prior mismatches in the context of posterior sampling using the Unadjusted Langevin Algorithm (ULA). Our work is novel since there is no prior work with such theory within ULA or even the Diffusion Models (DM) literature.
>
> Diffusion models (DM) and Unajusted Langevin Algorithm (ULA) are both
> based on a stochastic process simulation, but they have profound
> differences. These two methods do not rely on the same stochastic
> equation, so that their analysis is different and the obtained result is
> not straightforward. ULA has an unlimited number of iteration (which can
> be very large in practice) whereas DM requires a discretization of a
> finite interval. In this context, it is noticeable that our analysis of
> PnP-ULA provides a similar metric (posterior-$L_2$ pseudometric) that
> the one appearing in DM analysis.

---

> ### Author Response · Authors · 2023-11-19
> **Response to Reviewer rc8y (part 2/2)**
>
> - It seems like the theoretical results drawn rely on ``oracle" information that is unavailable in practice. So the practical use of this theoretical tool is largely unclear for me.
>
> We would like to point out that the main result in Theorem 1 does not rely on "oracle" information. The posterior-$L_2$ pseudo-metric and the Wasserstein distance between samplings are tractable in practice (see Fig.~1 in the main paper). We do not use any ``oracle" information to derive this bound.
>
> The practical implications are twofold: 1) Our results indicate that the posterior-$L_2$ distance is a better metric than the prior-$L_2$ distance (the $L_2$ distance computed on the training data) to compare different denoisers. This has been proven experimentally in the GMM experiment (Fig. 1 left plot). This gives a practical criterion to choose the most relevant denoiser for a given task. 2) Our Theorem 1 provides a stability guarantee of PnP-ULA in a realistic case of application (mismatched measurement model and mismatched denoiser). This stability serves as validation for the practical utility of the algorithm.

---

### Official Review · Reviewer_Uuqz · 2023-11-01

**Soundness:** 2 fair
**Presentation:** 2 fair
**Contribution:** 2 fair
**Rating:** 5
**Confidence:** 4

**Summary:**

This paper considers sensitivity analysis of posterior sampling in inverse problems using diffusion models. It analyzes the effects of mismatches to the drift function on the stationary distribution of Langevin sampling.The mismatch can arise due to uncertainty in the forward operator and due to the denoiser not being exactly the MMSE denoiser.

The main result is that the stationary distributions differ proportional to a pseudometric that depends on the drift mismatch.

**Strengths:**

The considered problem is relevant, especially in medical imaging, where we want algorithms to be robust to mismatch in the forward model.

Inverse problems using diffusion models is also an active area of research and the proposed results could be relevant.

**Weaknesses:**

- The main result in Theorem 1 shows that the TV between the stationary distributions of two Markov chains that have different drift functions can be bounded in terms of the proposed ``posterior-$L_2$ pseudometric''. This pseudometric is defined in terms of the expectation of the difference between the two drift functions when samples are drawn from the stationary distribution of one of the drifts. It's not clear at all how this pseudo metric behaves, and whether it is sufficiently small for two drifts that are close. (the $\epsilon$ used in the results is only for the mollification level present in the denoiser, and has nothing to do with the closesness of the drifts themselves).

- It is also not clear how different the two stationary distributions are when compared to the continuous stationary distribution. This can be very different due to discretization error, etc.

- There is very little comparison to existing results in the literature. Other than saying that their results are backwards compatible with Laumont et al 2022, the authors do not state what benefits / drawbacks their results face.

- The paper considers Langevin sampling, which is known to not mix very well -- most theoretical results in the literature consider ODE / SDE solvers for an Ornstein–Uhlenbeck process.

- The upper bounds in Theorem 1 are specified in terms of $A_0, A_1, B_0, B_1$, without any mention on the dimension dependence of these quantities.

- Some statements are unsubstantiated. In the contributions section, the authors claim "This paper stresses that in the case of mismatched operators, there are no error accumulations." I don't see why this would we be true.

**Questions:**

Listed in the weaknesses section

---

> ### Author Response · Authors · 2023-11-18
> **Response to Reviewer Uuqz (part 1/2)**
>
> Thank you for your positive feedback and careful reading of our paper. Below we provide detailed response to your comments.
>
> - The main result in Theorem 1 shows that the TV between the stationary distributions of two Markov chains that have different drift functions can be bounded in terms of the proposed ``posterior-pseudometric". This pseudometric is defined in terms of the expectation of the difference between the two drift functions when samples are drawn from the stationary distribution of one of the drifts. It's not clear at all how this pseudometric behaves, and whether it is sufficiently small for two drifts that are close. (the $\epsilon$ used in the results is only for the modification level present in the denoiser, and has nothing to do with the closesness of the drifts themselves).
>
> The posterior-$L_2$ pseudometric is the expectation of the $L_2$ norm with respect to the law of sampling $\pi_{\epsilon, \delta}^1$. We introduce this pseudometric to compute a distance between two drift functions. This pseudometric is by definition positive, symmetric, and verifies the triangular inequality. However, it is not a distance, since the separability is not satisfied ($d_1(b_{\epsilon}^1, b_{\epsilon}^2) = 0$ do not imply $b_{\epsilon}^1 = b_{\epsilon}^2$). For two drift functions $b^1$ and $b^2$, where $\|b^1 - b^2\|_{\infty} \le \epsilon$ (for example close forward-model or close denoisers), then the posterior-$L_2$ pseudometric $d_1(b^1, b^2) \le \epsilon$.
>
> Prompted by your comment, we have revised the description of the posterior-$L_2$ in Appendix A.
>
> - It is also not clear how different the two stationary distributions are when compared to the continuous stationary distribution. This can be very different due to discretization error, etc.
>
> This is a good point that we have considered in Corollary 1.2 in our submission, where we compare the error between the empirical stationary distribution and the posterior distribution with the noisy prior $p_{\epsilon}(\cdot|\mathbf{y})$. Three different terms bound the error between the empirical stationary distribution and the posterior distribution. The first is the drift shift due to the denoiser mismatch, the second is a discretization error, and the third is a projection error.
>
>  - There is very little comparison to existing results in the literature. Other than saying that their results are backwards compatible with Laumont et al 2022, the authors do not state what benefits / drawbacks their results face.
>
> We have compared our results with related theorems that use pseudometric $d_1$ to analyze diffusion models [1-3] after Theorem 1 in our initial submission.
>
> Prompted by your remark, we have strengthened both the benefits and drawbacks of our theoretical results in the revision. The main advantage of our theorem is its provision of a uniform mathematical formulation capable of capturing both the forward-model mismatch and the denoiser mismatch, which has not been investigated theoretically in the context of Bayesian imaging. Our theorem proves that PnP-ULA is a stable method with respect to a mismatched drift. This is a powerful theoretical guarantee as such shifts occur in practice.
>
> On the other hand, our Theorem 1 indeed has the drawback of having constants $A_0, B_0, A_1, B_1$ that depends on the observation, i.e., the target posterior distribution. These constants are not tractable in practice and depend on the dimension $d$. Such constants is a standard component in ULA analysis [4, 11]. We have analyzed these constants in section D.1.6 in the revised supplementary material.
>
>  - The paper considers Langevin sampling, which is known to not mix very well -- most theoretical results in the literature consider ODE / SDE solvers for an Ornstein–Uhlenbeck process.
>
> Unajusted Langevin Algorithms (ULA) and Ornstein–Uhlenbeck processes (such as Diffusion Models, DM) both rely on stochastic equation simulation, as explained in  [5, section 2.1].
> Thus the analysis of ULA and DM rely on similar mathematical tools, associated with Girsanov's theory.
> We underline that there is an extensive literature of ULA previous to the emerging of diffusion model and that many fundamental theoretical results are based on Langevin sampling (see for example [6-8]).
>
> Indeed ULA is known to mix slowly, whereas the annealing in the reverse Ornstein-Uhlenbeck process helps it explore modes more efficiently. That said, this is not a reason for discarding ULA altogether: ULA may be used in certain situations where diffusion models are not easy to apply. The texture synthesis algorithm in [9] is a notable example.
> In addition, despite slow mixing times, our experiments suggest that PnP-ULA has the ability to recover simple multimodality (see the different type of eyebrow on figure 9). We added a paragraph "Do we capture multimodality with PnP-ULA ?" in Appendix C.1 to discuss this point.

---

> ### Author Response · Authors · 2023-11-18
> **Response to Reviewer Uuqz (part 2/2)**
>
> - The upper bounds in Theorem 1 are specified in terms of $A_0, A_1, B_0, B_1$, without any mention on the dimension dependence of these quantities.
>
> The constant of the geometrical ergodicity $B_a$ (and thus of $A_0$, $B_0$, $A_1$, $B_1$) has a polynomial dimension dependence.
> Notice that the speed of the geometrical ergodicity $\rho_C$ [10, Definition 15.1.1] of the Markov Chain has been proved to be independent of the dimension by [4]. This is now stated in the revised version of the paper.
>
> We also added a paragraph (section D.1.6) at the end of the proof of Theorem 1 to detail the behaviour of each constant. The proof itself has also been completed to express $B_0$ explicitly.
>
>  - Some statements are unsubstantiated. In the contributions section, the authors claim "This paper stresses that in the case of mismatched operators, there are no error accumulations." I don't see why this would we be true.
>
> Note how Theorem 1 showes that a small shift in the drift (the deterministic term of the stochastic process) results in only a minor shift in the invariant law.
>
> The invariant law of this Markov Chain is the limit law of $\mathbf{x_k}$ as $k \to +\infty$. At each step $\mathbf{x_k} \to \mathbf{x_{k+1}}$, there is an error due to the drift shift. However, as $k \to +\infty$, errors at each step do not accumulate indefinitely, and they only imply a finite error on the invariant law (Theorem 1). Therefore, PnP-ULA is robust to a drift shift.
>
> Fundamentally, there is no exploding error because this Markov Chain is geometrical ergodic [10, Definition 15.1.1] as we prove in Eq.~(23). This geometrical ergodicity is a key argument of the proof of Theorem 1.
>
> In order to avoid possible ambiguities, we have reformulated this sentence as "This paper stresses that in the case of mismatched operators, errors do not accumulate indefinitely" in the revision.
>
> [1] Hongrui Chen, Holden Lee, and Jianfeng Lu. Improved analysis of score-based generative modeling:
> User-friendly bounds under minimal smoothness assumptions. In International Conference on
> Machine Learning, pp. 4735–4763. PMLR, 2023.
>
> [2] Joe Benton, Valentin De Bortoli, Arnaud Doucet, and George Deligiannidis. Linear convergence
> bounds for diffusion models via stochastic localization, 2023.
>
> [3] Giovanni Conforti, Alain Durmus, and Marta Gentiloni Silveri. Score diffusion models without
> early stopping: finite fisher information is all you need, 2023.
>
> [4] Valentin De Bortoli and Alain Durmus. Convergence of diffusions and their discretizations: from
> continuous to discrete processes and back, 2020.
>
> [5] Yang Song, Jascha Sohl-Dickstein, Diederik P Kingma, Abhishek Kumar, Stefano Ermon, and Ben
> Poole. Score-based generative modeling through stochastic differential equations. arXiv preprint
> arXiv:2011.13456, 2020
>
> [6] Gareth O. Roberts and Richard L. Tweedie. Exponential convergence of langevin distributions and
> their discrete approximations. Bernoulli, 2(4):341–363, 1996. ISSN 13507265.
>
> [7] Andreas Eberle. Reflection couplings and contraction rates for diffusions. Probability theory and
> related fields, 166:851–886, 2016.
>
> [8] Alain Durmus and Eric Moulines. High-dimensional bayesian inference via the unadjusted langevin
> algorithm, 2018.
>
> [9] Valentin De Bortoli, Agn`es Desolneux, Alain Durmus, Bruno Galerne, and Arthur Leclaire. Maxi-
> mum Entropy Methods for Texture Synthesis: Theory and Practice. SIAM Journal on Mathemat-
> ics of Data Science, 3(1):52–82, jan 2021. ISSN 2577-0187. doi: 10.1137/19M1307731.
>
> [10] Randal Douc, Eric Moulines, Pierre Priouret, and Philippe Soulier. Markov chains. Springer, 2018.
>
> [11] Xiang Cheng and Niladri S. Chatterji and Yasin Abbasi-Yadkori and Peter L. Bartlett and Michael I. Jordan, Sharp convergence rates for Langevin dynamics in the nonconvex setting, 2020

---

> ### Author Response · Authors · 2023-11-22
> **Comment to Reviewer Uuqz**
>
> Dear Reviewer, as we are nearing the end of the discussion period, please let us know if there are any other remaining questions that we could address.

---

### Official Review · Reviewer_eQV2 · 2023-11-06

**Soundness:** 4 excellent
**Presentation:** 2 fair
**Contribution:** 2 fair
**Rating:** 6
**Confidence:** 3

**Summary:**

The authors study the influence of model mismatch on the invariant distribution of a certain model-based posterior sampler based on the unadjusted Langevin Algorithm. The model (a forward operator in some imaging application) and the prior (incorporated via a denoiser) factor in through the drift term. The authors prove that the distribution drift is controlled by the total size of the model mismatch (in the forward model and the denoiser).

**Strengths:**

It is great that the paper proves a theoretical result about a relevant topic where most work is highly empirical. The setting is very clear and the derivations seem sound (although I have not checked in great detail.) The authors work under fairly general assumptions and also illustrate their bounds empirically.

**Weaknesses:**

- The contribution is somewhat incremental given all the preparatory work in Laumont et al. 2022. Model mismatch is certainly a relevant topic and it is nice to have a paper about it so this is a somewhat subjective statement relative to papers I've reviewed for ICLR this year.

- The prose is waffly, with too much hyperbole. An example: "In this section, our focal point resides in the investigation of the profound impact of a drift shift on the invariant distribution of the PnP-ULA Markov chain" which could be "In this section we study the impact of drift shift on...".

- There are also numerous typos and broken sentences, especially in the appendices.

**Questions:**

- Under "Contributions" you say that you "provide a more explicit re-evaluation of the previous convergence results...", but I am not sure what this means.

- In "we focus on the task of sampling the posterior distribution to reconstruct various solutions...", what is meant by "various solutions"?

- In "... Markov chain can be naturally obtained from an Euler-Maruyama discretisation by reformulating the process ...", what is meant by "reformulating"?

- Before equation (7), Wasserstein norm should be Wasserstein metric (or distance); before (7), TV distance should be TV norm (which is what is defined in (7)). (Also: why are Rd vectors bold in (8) and not in (7)?)

- "pseudometric between two functions in Rd" -> taking values in Rd

- In Corollary 1.3 which norm is || A^1 - A^2 ||?

---

> ### Author Response · Authors · 2023-11-16
> **Response to Reviewer eQV2 (part 1/2)**
>
> Thank you for your positive and accurate feedback, and the careful reading of our paper. We answer to the comments and questions of the reviewer in details.
>
> - The contribution is somewhat incremental given all the preparatory work in Laumont et al. 2022. Model mismatch is certainly a relevant topic and it is nice to have a paper about it so this is a somewhat subjective statement relative to papers I've reviewed for ICLR this year.
>
> As noticed by the reviewer, model mismatch is a key issue to study in order to use Bayesian methods in practice, and our work gives new theoretical perspectives about these mismatches.
> Our paper is indeed based on the method and the previous analysis in Laumont et al. 2022. There is nevertheless a main theoretical difference as we get rid of any assumption on the distance between the mismatched MMSE denoiser and the exact MMSE denoiser. In  H2 of Laumont et al. 2022, this distance is assumed to be bounded by a constant. Our work is therefore the first one to explore PnP-ULA in a perspective of mismatched denoisers and mismatched measurement model.
>
>  - The prose is waffly, with too much hyperbole. An example: "In this section, our focal point resides in the investigation of the profound impact of a drift shift on the invariant distribution of the PnP-ULA Markov chain" which could be "In this section we study the impact of drift shift on...".
>  - There are also numerous typos and broken sentences, especially in the appendices.
>
> Thank you for your remark. We simplified the prose and adderessed some errors. In particular, in the proof (Appendix D) we added an introductory paragraph that provides a global overview of the different parts of the proof and how they are combined. We hope this helps readability.
>
> - Under "Contributions" you say that you "provide a more explicit re-evaluation of the previous convergence results...", but I am not sure what this means.
>
> Corrolary 1.2 is a new bound for the main convergence result of Proposition 6 of Laumont et al. 2022. This bound expresses the shift between the stationary distribution of the Markov Chain $\pi_{\epsilon, \delta}^2$ and the posterior distribution with a noisy prior $p_{\epsilon}(\cdot,\mathbf{y})$. This shift is bounded by the discretization error, the projection error and the posterior-$L_2$ pseudometric. This bound is claimed as to be more explicit because we use the posterior-$L_2$ pseudometric to measure the distance between $D_{\epsilon}$ and $D_{\epsilon}^\star$ while H2 of Laumont et al. 2022 assumes a constant.
>
> To better address your comment, the second contribution has been reformulated as follows: "Furthermore, we provide in Corollary 1.2 a generalized convergence result for PnP-ULA (Laumont et al. 2022). A new proof strategy based on Girsanov's theory allows us to relax assumptions on the denoiser accuracy."
>
> - In "we focus on the task of sampling the posterior distribution to reconstruct various solutions...", what is meant by "various solutions"?
>
> Ill-posed inverse problems in imaging may admit several plausible solutions. This is typically the case when the posterior distribution is  multimodal (see Figure 8 in the supplementary material). Hence, our objective is to capture this diversity by considering a sampling method (here PnP-ULA) that is able to generate multiple samples from the posterior distribution (i.e. various solutions for the tackled problem).
>
> We propose to reformulate the sentence as "A posterior sampling approach produces multiple solutions for the inverse problem reconstruction." Moreover a detailed discussion has also been added in Appendix C.1.

---

> ### Author Response · Authors · 2023-11-16
> **Response to Reviewer eQV2 (part 2/2)**
>
> - In "... Markov chain can be naturally obtained from an Euler-Maruyama discretisation by reformulating the process ...", what is meant by "reformulating"?
>
> By the term "reformulating", we wanted to express the distinction between the continuous stochastic equation (Equation 1 in the main paper) and the discretization of this equation following the Euler-Maruyama scheme.
> Thank you for noticing the ambiguity of this formulation. We propose to reformulate this sentence as
> "In practice, an Euler-Maruyama discretization of Equation (1) defines the Unadjusted Langevin algorithm (ULA) Markov chain for all $k \in \mathbb{N}$ as...".
>
>  - Before equation (7), Wasserstein norm should be Wasserstein metric (or distance); before (7), TV distance should be TV norm (which is what is defined in (7)). (Also: why are Rd vectors bold in (8) and not in (7)?)
>  - "pseudometric between two functions in Rd" = taking values in Rd
>
> Thank you for noticing these typos that have been fixed in the revised version. Equation (7) has been modified. To clarify the distinction between TV norm and TV distance, the sentence before (7) has been modified as : "The $TV$ distance is the distance induced by the $TV$ norm defined for a probability density $\pi$ by". The term "Wasserstein norm" has been replaced with "Wasserstein distance". The remark on the pseudometric between functions has been taken in account.
>
>  - In Corollary 1.3 which norm is $|| A^1 - A^2 ||$?
>
> The norm used in Corollary 1.3 is the spectral norm. However, this bound is true for any norm on $\mathbf{R}^d$ because all norms are equivalent in finite-dimensional space. We propose to add the sentence "The spectral norm is used here, but this result holds true for any norm in the matrix space since all norms are equivalent in finite-dimensional space." after Corrollary 1.3.

---

> > ### Comment · Reviewer_eQV2 · 2023-11-22
> >
> > I thank the authors for the clarifications. As I wrote in the initial assessment, I think this paper addresses a relevant problem. The results build on earlier work and remove some of its limitations. I will thus retain my original score.

---

> > > ### Author Response · Authors · 2023-11-22
> > > **Response to Reviewer eQV2**
> > >
> > > Thank you again for reading our paper and our responses. We also appreciate the positive comments.

---

### Author Response · Authors · 2023-11-16
**Response to all reviewers**

We thank all the reviewers for their careful reading and all their remarks and questions. We update the paper file with a new document containing some modifications. All significant modifications are highlighted in blue. In particular, the proof has been clarified with additional paragraphs.

Our contribution are :

(a) Bayesian posterior sampling relies on two operators: a data-fidelity term and a denoising term. This paper stresses that in the case of mismatched operators, errors do not accumulate indefinitely. Moreover, with mismatched operators, the shift in the sampling distribution can be quantified by a unified formulation, as presented in Theorem 1.

(b) Furthermore, we provide a generalized convergence result for PnP-ULA (Laumont et al., 2022) in Corollary 1.2. A new proof strategy based on Girsanov theory allows us to relax assumptions on the denoiser accuracy. These insights are substantiated by a series of experiments conducted on both a 2D Gaussian Mixture prior and image deblurring scenarios.

---

### Comment · Area_Chair_p5cj · 2023-11-22

Dear all,

The author-reviewer discussion period is about to end.

@authors: If not done already, please respond to the comments or questions reviewers may further have. Remain short and to the point.

@reviewers: Please read the author's responses and ask any further questions you may have. To facilitate the decision by the end of the process, please also acknowledge that you have read the responses and indicate whether you want to update your evaluation.

You can update your evaluation positively (if you are satisfied with the responses) or negatively (if you are not satisfied with the responses or share other reviewers' concerns). Please note that major changes are a reason for rejection.

You can also keep your evaluation unchanged. In this case, please indicate that you have read the responses, that you do not have any further comments and that you keep your evaluation unchanged.

Best regards,
The AC

---

### Meta-Review · Area_Chair_p5cj · 2023-12-10

**Metareview:**

The paper has received borderline recommendations, but the reviewers lean towards acceptance (6-5-6-6). The paper is a theoretical study of the plug-and-play unadjusted Langevin algorithm  (PnP-ULA) for solving inverse problems. It shows that the sensitivity of the sampling distribution of PnP-ULA to a mismatch in the measurement model and the denoiser can be precisely characterized.

The authors-reviewers discussion has been somehwat constructive and has led to some improvements to the paper, in particular regarding its presentation and clarity. However, the reviewers have not changed their original recommendations.

Overall, I believe the paper presents an interesting contribution, which sheds some light on the actual behaviour of PnP-ULA. To me, the paper is of sufficient quality to be accepted. I recommend acceptance. I also encourage the authors to address the remaining concerns of the reviewers in the final version of the paper.

**Justification For Why Not Higher Score:**

Borderline scores

**Justification For Why Not Lower Score:**

Scores are all borderline positive, except for one, but the reviewer did not engage after the authors reply.

---

### Decision · Program_Chairs · 2024-01-16

Accept (poster)